# Triboelectric nanogenerator sensors for soft robotics aiming at digital twin applications

Tao Jin [1,2,3,7], Zhongda Sun [2,3,7], Long Li [1,7], Quan Zhang[1], Minglu Zhu[2,3,4], Zixuan Zhang[2,3], Guangjie Yuan[1], Tao Chen[5], Yingzhong Tian[1✉], Xuyan Hou [6✉] & Chengkuo Lee [2,3,4✉]

Designing efficient sensors for soft robotics aiming at human machine interaction remains a challenge. Here, we report a smart soft-robotic gripper system based on triboelectric nanogenerator sensors to capture the continuous motion and tactile information for soft gripper. With the special distributed electrodes, the tactile sensor can perceive the contact position and area of external stimuli. The gear-based length sensor with a stretchable strip allows the continuous detection of elongation via the sequential contact of each tooth. The triboelectric sensory information collected during the operation of soft gripper is further trained by support vector machine algorithm to identify diverse objects with an accuracy of 98.1%. Demonstration of digital twin applications, which show the object identification and duplicate robotic manipulation in virtual environment according to the real-time operation of the soft-robotic gripper system, is successfully created for virtual assembly lines and unmanned warehouse applications.

[1] Shanghai Key Laboratory of Intelligent Manufacturing and Robotics, School of Mechatronic Engineering and Automation, Shanghai University, Shanghai 200444, China. [2] Department of Electrical and Computer Engineering, National University of Singapore, 4 Engineering Drive 3, Singapore 117576, Singapore. [3] Center for Sensors and MEMS, National University of Singapore, 4 Engineering Drive 3, Singapore 117576, Singapore. [4] NUS Suzhou Research Institute (NUSRI), Suzhou 215123, China. [5] Jiangsu Provincial Key Laboratory of Advanced Robotics, School of Mechanical and Electric Engineering, Soochow University, Suzhou 215123, China. [6] State Key Laboratory of Robot Technology and System, Harbin Institute of Technology, Harbin 150080, China. [7]These authors contributed equally: Tao Jin, Zhongda Sun, Long Li. ✉email: troytian@shu.edu.cn; houxuyan@hit.edu.cn; elelc@nus.edu.sg

With the establishment of 5 G technology in the next few years, the cost of massive data transmission via wireless network will be much cheaper. Such infrastructure will enable the smart homes and lean/smart manufacturing by deploying huge amounts of sensors under the internet of things (IoTs) framework to realize real-time sensory information collection, data management and analysis[1]. In the future, the combination of artificial intelligence (AI) technology with the IoT technology will lead to AI of things (AIoT)-based living, working, and manufacturing environment, which provides efficient IoT operations, improved human–machine interactions and better capability of decision-making with respect to a complicated and dynamic system[2,3]. With the aid of AIoT, computation capacity, and ubiquitous sensory information, digital twin is proposed to be a digital copy of the physical system, i.e., a cyber-physical system, to perform real-time control and optimization of products and production lines, because the required time of getting an optimized solution from cloud server reduces to an ignorable level[4]. One of the digital twin applications is the components sorting and complex product assembly at the shop floor. However, due to the complexity of building a digital equivalent in cyberspace to its physical counterpart, very limited progress has been achieved. So far, the visual (or image) recognition is the main approach to collect information for creation of real-time twin information in the digital space, i.e., the digital twin. We need to have sensors built-in[5] in the robotic manipulators, to recognize shape and size (or even type) of objects (or components) for sorting-assembly lines in an unmanned factory, because visual recognition[6] does not work in dark space, which is the trend for the realization of unmanned factory for the sake of energy saving.

Robotic manipulators have been developed for years, owing to their great assistance in various industries, and, hence, different designs of the robotic wrists were developed to enable those robots to accomplish the specific tasks[7]. In the meantime, the corresponding sensors were applied to monitor their motions and the external stimulus as well, especially for those surgical or micro-nano manipulation fields, the accurate motion detection and tactile sensing become quite important[8–12]. As for soft robotics, featuring high compliance and dexterity with muscle-like actuators, soft materials including silicone rubber and thermoplastic polyurethanes (TPUs) have been widely used in the fabrication[13–16]. Hence, the high nonlinear deformation and no-joint structure hinder the application of the traditional sensors such as potentiometer and encoder, causing the necessity of sensors compatible for soft robots[17]. For instance of sensor topology, the embedded soft resistive sensors were utilized to capture the deformations of soft grippers together with vision-based motion capture system[18]. Zhao et al.[19] have proposed a stretchable optical waveguide strain sensor, which was directly integrated into a soft prosthetic hand that can sense the shape and stiffness of contacted objects. Based on triboelectric mechanism, Lai et al.[20] have developed a self-powered stretchable robotic skin to help soft robot sense and interact with environment via self-generated signals, indicating another solution that has high compatibility with soft robots. Till now, the proposed solutions to achieve perception for soft robotics include photo detection[19,21], triboelectricity[22,23], piezoelectricity[24], electromagnetic effect[25], and conductive nanocomposites[18,26]. However, some of the above methods are restrained by several drawbacks, e.g., the electromagnetic field is required for Hall sensor[25] and nonlinearity or creep is a material concern in the nanocomposites[18]. The capability of detecting the continuous motion and external stimuli of the soft robot remains as grand challenges in soft robotics.

Noticeably, triboelectric nanogenerator (TENG)[27–30]-based sensors are more compatible with soft robotics, because the Young's modulus of soft materials typically used in triboelectric sensors is in the same level as the silicone rubber and TPU. More importantly, self-generated sensor output in response to the strains and deformation of TENG makes the design of signal processing circuits more straightforward. Besides, TENG has been investigated as promising solutions to harvest energy from body motion and save collected energy in capacitors for wireless sensor signal transmission[31], or to perform direct wireless sensory information transmission for IoT applications[32]. Therefore, using TENG approaches to realize tactile sensing skins integrated with soft or rigid actuators have been reported by Professor Wang's group frequently[20,33].

However, these studies only address contact-separation detection, without the capability of detecting the sliding and continuous motions, which requires further study. Recently, glove-like human–machine interfaces (HMIs) based on the stretchable and flexible sensors using resistive, capacitive, piezoelectric, or triboelectric mechanisms have been investigated frequently[34–37]. For instance, a bending sensor for continuous finger motion measurement can be achieved by TENG with grid pattern[38]. Meanwhile, there is also a trend of analyzing the massive data collected from a large amount of distributed sensors. Sundaram et al.[39] reported a glove with a dense matrix of 548 resistive sensors, to obtain the grasping signatures. In contrast, another approach is to leverage the minimalistic design, i.e., the minimum number of sensors, to provide just enough sensory information as a low-cost solution requiring less computation capacity[32]. To provide a human-like perception for robots, Boutry et al.[40] proposed a multi-dimensional capacitive sensor deployed on robotic finger, to identify both normal and shear forces with high sensitivity, and this kind of researches illustrate accurate dexterity of hand information, which may benefit the design for the sensory system of soft robotics. In addition, to further explore the potentials of these facile designed TENG sensors, the machine learning (ML) technique can be utilized to enhance the data interpretation for better manipulation or detection, i.e., accurate gesture recognition, which is equivalent to the continuous motion sensors[41]. However, there is little research reporting the TENG sensor-integrated soft pneumatic finger with ML-assisted recognition in details, while the significance of relevant technique is already addressed in the reported works, such as self-powered stretchable soft-robotic skin as mentioned previously[20].

Herein, TENG sensors composed of patterned-electrode tactile sensors and gear-structured length sensors are proposed for enhancing the intelligence of the soft manipulator (i.e., soft gripper), as shown in Fig. 1. The developed sensing system with patterned-electrode tactile TENG (T-TENG) sensor can detect sliding, contact position, and gripping mode of the soft gripper (see comparison to other methods in Supplementary Table 1). The measurement of bending angle of the soft actuators by length TENG (L-TENG) sensor generates signals by the contact separation between the electropositive gear teeth and the electronegative materials. Therefore, both of the motion caused by self-actuation and external stimuli can be sensed. A glove-based HMI composed of a single-electrode tactile sensor and an L-TENG sensor is used to perform the real-time control of the robotic hand to verify the real-time signal processing system. Next, to realize the feedback function, a tri-actuator soft gripper is fabricated by three-dimensional (3D) printing with integrated TENG sensors. By leveraging the ML technology for data analysis, the gripper is successfully demonstrated to perceive the gripping status and realize object identification, and a demonstration of digital twin is then established to create duplicate digital information of the above manipulation in virtual reality (VR) environment, i.e., the cyberspace. In general, the proposed devices together with soft gripper illustrated a great potential for digital

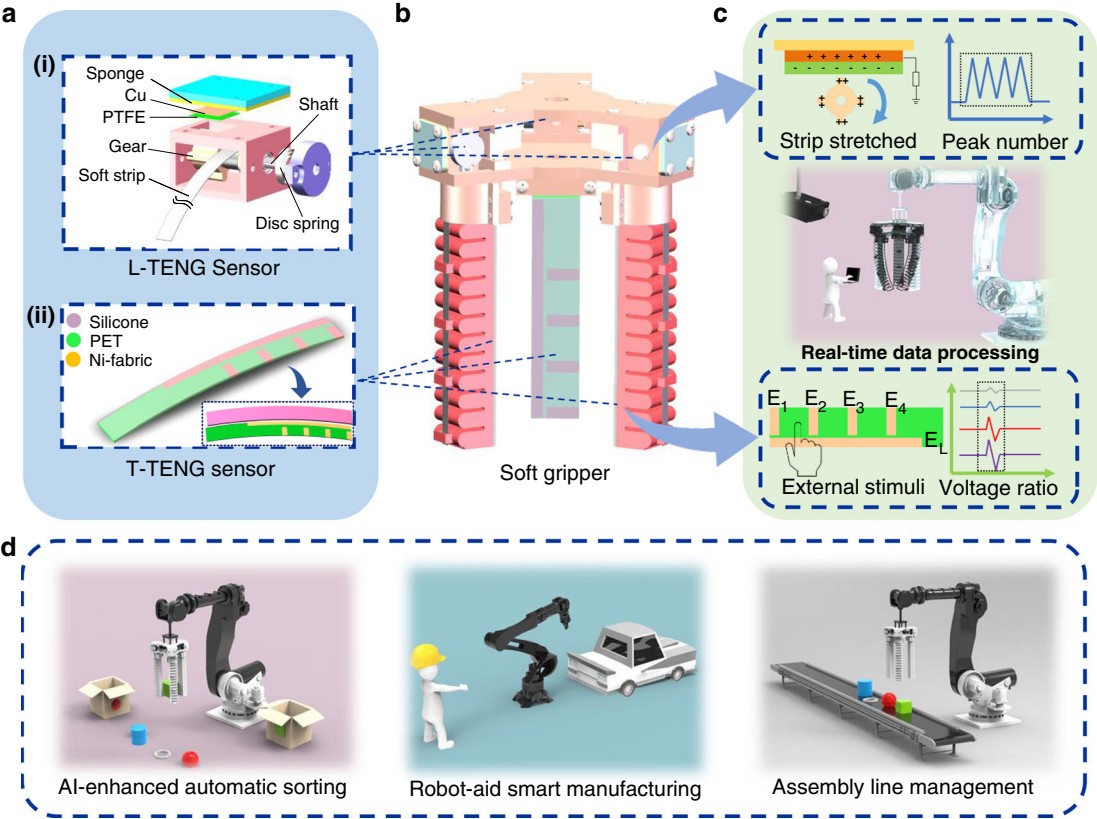

**Fig. 1 Construction drawing of the low-cost triboelectric nanogenerator (TENG) for soft gripper and its digital twin applications. a** The as-fabricated TENG sensors and their basic structures. (i) The length TENG (L-TENG) sensor and (ii) the tactile TENG (T-TENG) sensor. **b** The soft gripper integrated with TENG sensors. **c** The intelligent sensory data processing strategies. $E_1$ to $E_4$, and $E_L$ represent the electrodes in the T-TENG sensor. **d** The digital twin applications based on artificial intelligence of things (AIoT) sensory system.

twin applications, e.g., production management and prediction of situation in smart factory.

## Results

**Configuration and working mechanism of the L-TENG and T-TENG sensors.** Both of the L-TENG sensor and T-TENG sensor work in single-electrode mode. As depicted in Fig. 1a(i), the gear coated with Nickel-fabric conductive textile (i.e., Ni-fabric) serves as the positive triboelectric component in L-TENG sensor, and the polytetrafluoroethylene (PTFE) film (10 mm × 10 mm) serves as the negative triboelectric layer with Cu electrode on the back side connected to the external output. The gear will be driven to rotate when the soft strip is stretched, resulting in the intermittent contact between PTFE layer and gear's teeth, thus generating the triboelectric output. Meanwhile, the spring mounted on the rotation shaft deforms during stretching, then releases and provides the pull-back force when the strip is going to contract. This self-recovery ability enables the strip to remain tensed and avoids the deviation caused by the buckling of the thin strip. The structure of T-TENG sensor is shown in Fig. 1a(ii), five electrodes made of Ni-fabric are patterned on the polyethylene terephthalate (PET) substrate where all the electrodes are 5 mm wide and the short electrodes are equidistantly arranged with 20 mm intervals. Moreover, the long electrode is located along the length direction of the T-TENG sensor patch. With a layer of silicone rubber, a kind of flexible and stretchable negative triboelectric material, coated on the surface of the patch, triboelectric signals can be generated by the stimuli on the T-TENG sensor's surface and collected in these five electrodes. The detailed methods for the fabrication of T-TENG sensor can be found in

the "Methods" section and Supplementary Fig. 1. In addition, the proposed sensors can be directly applied to various types of robots, including those rigid materials. Leveraging ML technology, the soft gripper (Fig. 1b) composed of these two TENG sensors is successfully demonstrated to perceive the gripping status and realize object identification, and a demonstration of digital twin is then established to create duplicate digital information of the above manipulation in virtual environment. The signal readout strategies (Fig. 1c), including peak counting and voltage ratio, eliminate the issues of absolute amplitude fluctuation caused by the variations of environmental factors, such as humidity and temperature. The further improvement of sensing resolution can be achieved via precise fabrication process such as metallurgy and machining. Hence, the proposed digital twin system shows its great potential in the area of unmanned warehouse, smart factory, intelligent assembly line, and IoT as shown in Fig. 1d.

The working mechanism of the T-TENG sensor is illustrated in Fig. 2a, b. The short electrodes in T-TENG sensor are marked as $E_1$ to $E_4$ and the long electrode is marked as $E_L$. When external stimuli occur, due to triboelectrification and electrostatic induction effect, the electric potential between electrodes and ground changes, driving electrons to flow through external resistance to electrodes, thus generating a positive peak shown in Fig. 2a(i). Similarly, when stimuli are released, the electric potential will change back to its original state and the induced negative charges in electrodes will also flow back to ground and generate a negative peak. Due to the distribution of four short electrodes along the T-TENG sensor patch, the amount of induced charges in different electrodes differs according to the distances between electrodes

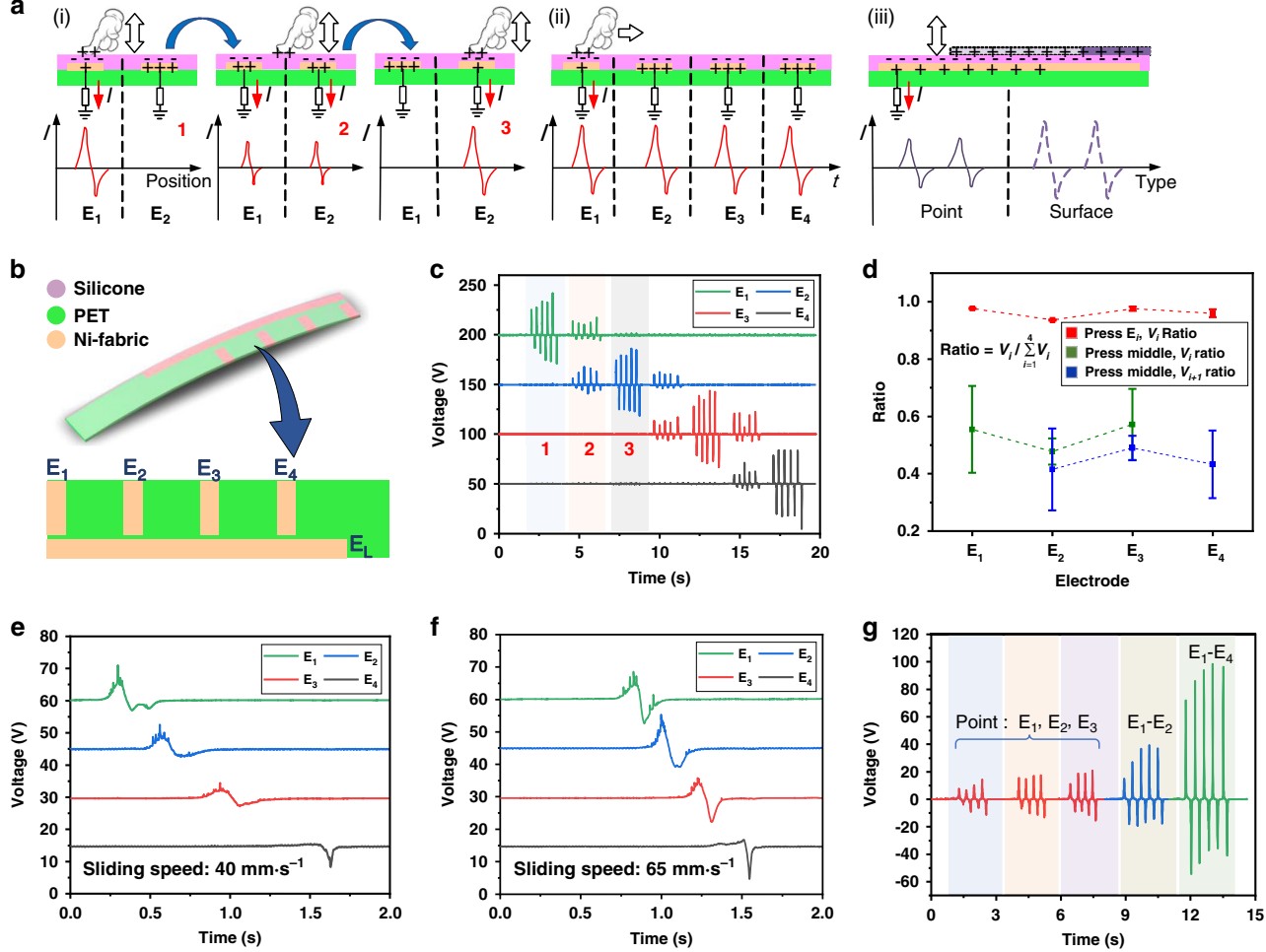

**Fig. 2 Working mechanism and characterization of the tactile TENG (T-TENG) sensor. a** Operation mechanism of the T-TENG sensor used for various functions including (i) contact position detection, (ii) sliding detection, and (iii) contact mode recognition. **b** Schematic diagram of the distribution of the electrodes. The short electrodes are marked as $E_1$ to $E_4$ and the long electrode is marked as $E_L$. **c** Output signals of $E_1$ to $E_4$ by tapping different positions across the T-TENG sensor patch in point-contact mode. **d** Voltage ratio analysis of the result in **c**, where $V_i$ means the voltage of the $i$th electrode. Sliding test at the speed of **e** 40 and **f** 65 mm·s$^{-1}$. **g** Output signals of $E_L$ when tapped with different contact areas.

and the contact point. The correspondence between contact position and the actual output can be seen from Fig. 2a(i), c (marked as 1, 2, 3). Based on this phenomenon, the output differences can be utilized to determine the specific position of the contact point by calculating the voltage ratio of these four short electrodes (Supplementary Fig. 2 and Supplementary Note 1). As for the sliding mode, the whole process can be seen as a combination of multiple stimuli shown in Fig. 2a(ii). Peaks are generated in sequence along the finger sliding direction and telling the continuous contact position in time domain. In addition, the long electrode $E_L$ is designed for detecting the total contact area, as well as perceiving gripper's gripping mode (point contact or surface contact) for the applications shown in Fig. 1d. As plotted in Fig. 2a(iii), as the contact area increases, the output will also be increased. Benefitting from the soft, flexible, and thin features of this silicone-based TENG sensor, the multifunctional T-TENG sensor can be quite compatible with the soft gripper serving as the touch sensing unit as illustrated in Fig. 1b.

To verify the output of the T-TENG sensor, a thin TENG patch (25 mm × 110 mm) is fabricated with three layers as shown in Fig. 2b. The load voltage generated by the finger tapping on different positions of this T-TENG sensor is shown in Fig. 2c. In this process, the finger taps both the positions above the four short electrodes and the middle area of two adjacent electrodes.

When tapping right above the electrode, most of the charges will be induced in the corresponding electrode comparing with other three electrodes. However, when tapping the middle area of two adjacent electrodes, both electrodes will have output, where the amplitudes are related to the distances between the finger tapping position and electrodes (Supplementary Fig. 2 and Supplementary Note 1). To further analyze the capability of discrimination, the output ratio of these four short electrodes, which can be listed as Ratio = $V_i / \sum_{i=1}^{4} V_i$ is calculated as shown in Fig. 2d, where $V_i$ means the output of $i$th electrode. It can be observed that the ratio remains high (over 93%) when tapping directly above the electrode ($E_1$ to $E_4$). While for tapping the middle positions, the voltage ratio fluctuates between 30% and 70%, which should be considered as a reasonable result considering its minimum detecting resolution of 10 mm with electrodes arranged in 20 mm intervals. In general, this approach bypasses the environmental noises, which may affect the output amplitude. Figure 2e, f show the output voltage with finger sliding across four electrodes in normal speed and fast speed, respectively. In this mode, the sliding speed can be calculated according to the time intervals between adjacent peaks, where a shorter time interval means faster sliding speed. The long electrode ($E_L$) located along the length direction of the T-TENG sensor is to detect the contact area between the soft-robotic finger and gripped objects (Fig. 2g).

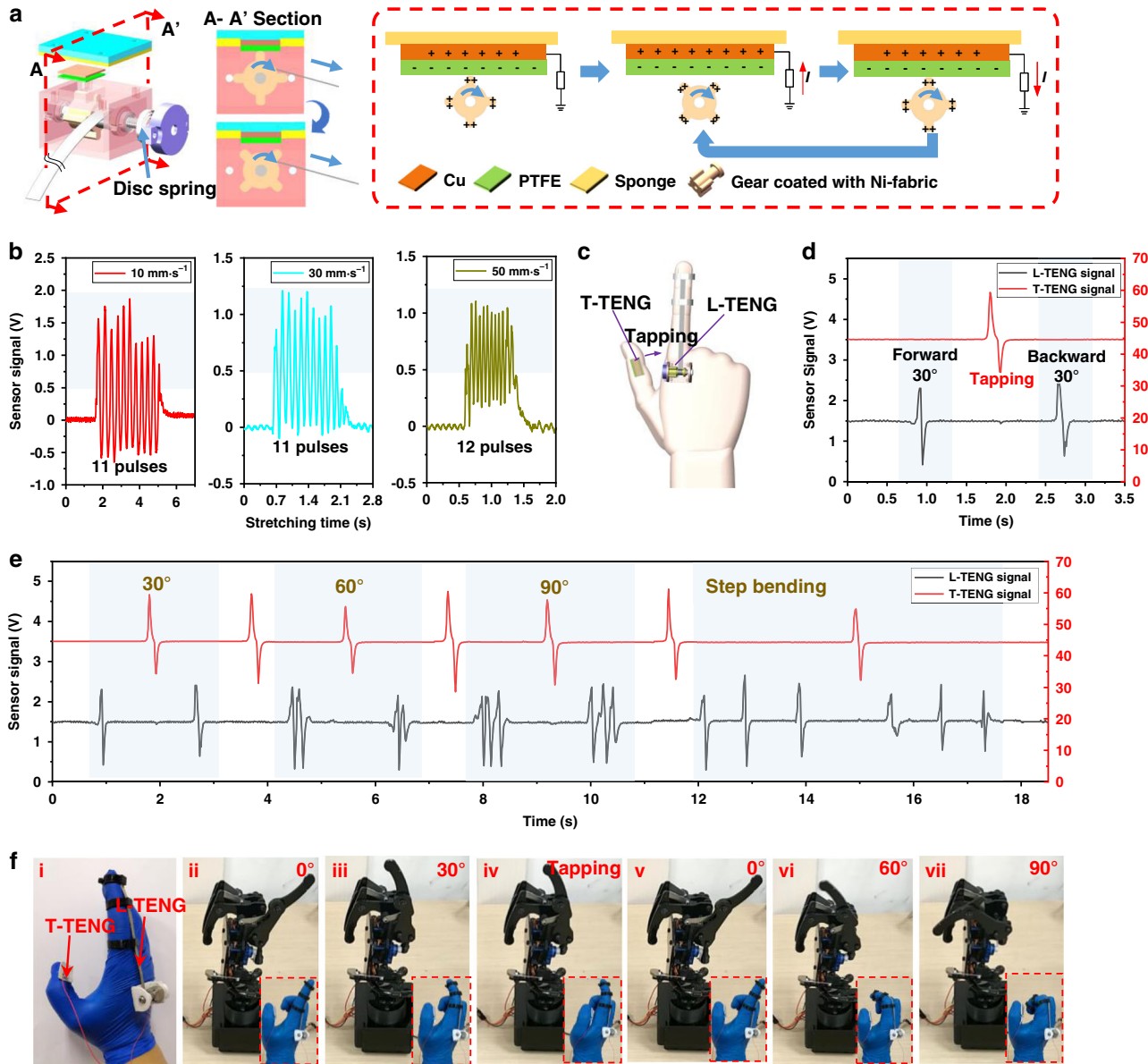

**Fig. 3 Working mechanism of the length TENG (L-TENG) sensor and the real-time verification in HMIs. a** Working mechanism of L-TENG sensor. **b** Stretching output of the L-TENG sensor at various speeds. **c** The photograph of the HMIs with L-TENG sensor and tactile TENG (T-TENG) sensor integrated on a nitrile glove. **d** The real-time signals when the robotic hand is controlled to bend to 30° and return to its original position. **e** The real-time signals of controlling the robotic finger to bend to different angles in one-step mode and step-by-step mode. **f** The photographs of the actual hand gestures and their corresponding robotic hand motions. (i) The nitrile glove with sensors. (ii–v) The robotic hand deforms to 30° and returns to 0°. (vi, vii) The robotic hand with 60°, 90° bending.

Here, we tap on the $E_L$ at the position of $E_1$, $E_2$, and $E_3$ in the length direction with the same force by finger. The outputs of these three taps generated in $E_L$ are almost the same due to the nearly equal contact areas. However, when the contact area increases from point area to a larger area, which can cover $E_1$ to $E_2$ (20 mm) or even $E_1$ to $E_4$ (60 mm), the output increases to be twice or four times larger than that of the point-contact condition. This output variation shows the possibility of perceiving the gripping mode (i.e., point contact or surface contact) of the soft gripper.

The working mechanism of the L-TENG sensor is shown in Fig. 3a where a four teeth gear is used. The L-TENG sensor is made of a PTFE layer with a copper electrode and a gear covered with Ni-fabric, which can induce the pulse signals as the gear rotates. First, when the teeth start contacting the PTFE layer, the

PTFE layer tends to attract electrons that can be retained on its surface due to the difference of electron affinity, while the Ni-fabric layer tends to lose the electrons, thus producing tribo-electric static charges on the contact surface. Then, as the gear is driven to rotate by the stretching soft strip, the intermittent contacts between PTFE layer and gear's teeth will result in the continuous alternating of electrical potential between Cu electrode and the ground, driving electron flow and generating the cyclic output peaks. To explore the performance of L-TENG sensor in various stretching situations, a programmable linear stepper motor as shown in Supplementary Fig. 3 is applied to control strip's motion and the open-circuit tests can be seen in Supplementary Movie 1. The maximum stretchable length of the strip is 60 mm, which is limited by the disc spring mounted on the rotation shaft that provides recovery force, so the L-TENG

sensor here is tested with a 55 mm stroke. The strip is controlled to be stretched and released by the stepper motor at a speed of 10, 30, and 50 mm·s$^{-1}$, respectively, and the signals generated during stretching process can be seen in Fig. 3b. For a lower threshold of 0.5 V, the corresponding output peak numbers of the stretching processes are 11, 11, and 12, respectively, showing the stability of length detection with acceptable small error, whereas the peak numbers for recovery process are 10, 10, and 11, respectively (Supplementary Fig. 4). Although the movement distances are the same, the shortage of one peak for recovery process mainly comes from the reverse motion, which results in corresponding inverse waveform and the number of negative peaks in recovery process is the same as the positive peaks in stretching process. This phenomenon could be used to distinguish the motion direction of the strip in different operation process. In brief, this L-TENG sensor can detect the displacement with the minimum resolution of 5 mm with an acceptable error rate of 8.3% when the external load is small (Supplementary Fig. 5). In addition, the resolution can still be further improved by decreasing the diameter of the shaft connected with the driven strip, increasing the number of the gear teeth, etc.

For verifying the real-time output of this L-TENG sensor and its compatibility with soft robotics, the L-TENG sensor is mounted on a nitrile glove with three TPU-based rings fixed on the index finger to guide the soft strip. A 100 MΩ external resistance is connected with the sensor to match the internal impedance of triboelectric material. As depicted in Supplementary Fig. 4 and Supplementary Movie 2, when finger bends and stretches the soft strip, the real-time signals generated from the rotating gear can be directly detected by the oscilloscope. The signals for both the stretching and recovery process under different bending degrees are collected and the result shows that the bending angles can be clearly and easily recognized according to the peak numbers of the triboelectric output. In general, the L-TENG sensor illustrates a promising sensing capability that can be applied to continuous motions' real-time monitoring by counting the output peak numbers, showing its great potential in various motion perceiving applications such as gait monitoring and limb bending detecting, etc.

**Real-time control verification and robotic applications**. To demonstrate the real-time control capability of the T-TENG and L-TENG-sensing system, a glove-based HMI integrated with this sensing system is developed to control a robotic hand. As depicted in Fig. 3c, the L-TENG sensor is mounted on the index finger to control robotic finger's bending motion, and another tactile sensor located on the thumb is a small segment diced from the original T-TENG sensor to switch the bending direction of robotic finger by tapping. The schematic diagram of robotic control is shown in Supplementary Fig. 4, where an analog-to-digital chip (ADC) connected with a signal processing circuit receives data from the sensors and transmits the signal to control the robotic hand. As illustrated in Fig. 3d, the signal of L-TENG sensor is tested with a small bending degree where one peak (marked in black) of the L-TENG sensor's output here can be used to control the robotic finger to bend 30°, and the tapping signal (marked in red) generated by T-TENG sensor can change the bending direction of the robotic finger. The real-time signals in the left half of the Fig. 3e show a continuous one-step process of robot cooperative control, where the integrated sensing system captures real finger's bending motion and reflects to the robotic finger simultaneously. For a 30° bending of the robotic finger as shown in Fig. 3f(ii-iii), one peak (marked in black) is generated to trigger the downward bending of robotic finger. To switch the bending direction of robotic finger, a peak (marked in red) is then

generated by making the thumb tap on the index finger, as depicted in Fig. 3f(iv). Hence, the second black peak appears as the human finger bends up 30° to return the robotic finger to its original position (0°) (Fig. 3f(v)). Similarly, the bending of 60° and 90° is also demonstrated with the peak number of triboelectric outputs from L-TENG sensor for defining the bending degree of one-step bending motion (two peaks for 60° and three peaks for 90°). Moreover, the robotic finger is also demonstrated with the sequential bending (30° for each step) from the original position (0°) to 90° as the signals in the right half of Fig. 3e. The hand gestures and motions of the robotic hand according to the real-time signals in Fig. 3e are shown in Fig. 3f (also see Supplementary Movie 3). This self-powered HMI integrated with our well-designed L-TENG and T-TENG sensor show its great ability in continuous human motion monitoring and robot control, and thus have the potential to be used in the real-time perception of the soft gripper.

Soft actuator with one degree of freedom (DOF) plays an important role in the soft-robotic area. Crawling robots can be formed by a single soft actuator, while the tri-actuator can be further designed as a soft gripper or a soft-robotic arm. Here, the working mechanism of the soft actuator with a hollow-bellows structure is shown in Fig. 4a. The upper surface of the actuator is formed in a corrugated structure with lower stiffness and higher stretchability. Meanwhile, the lower surface consists of the same material without corrugated structure, which is relatively harder to deform. Hence, when the soft actuator is inflated, the upper surface tends to perform larger deformation, resulting in the bending down motion of the soft actuator driven by the air pressure difference between the chamber and the external environment, as shown in Fig. 4a. Besides, the increasing air pressure enlarges the bulge of the soft material and the sidewalls of the corrugated surface contact to enhance the output force as plotted in Fig. 4a. Consequently, the bending degree is related to the length difference between the lower length $L_1$ and the upper length $L_2$. Referring to related previous works[15,21], $L_1$ can be considered as a constant value and the bending degree is able to be calculated based on the constant-curvature theory where the shape of the deformed actuator can be seen as an arc of a circle. Therefore, the calculation equation can be listed as $\theta = (L_2 - L_1)w^{-1}$, where $w$ is the distance between the upper and lower surface curve, and the parameter $L_2$ here can be used for closed-loop control of the soft actuators.

To explore the compatibility of the L-TENG sensor in soft-robotic applications, a soft actuator integrated with the L-TNEG sensor is prepared with a pre-bending degree of 40° to enhance the recovery ability and enlarge the maximum bendable angle. The main structure of the soft actuator and the fabrication method are shown in Supplementary Figs. 6 and 7. As shown in Fig. 4b, the soft strip of L-TENG sensor is guided by rings on the actuator surface and fixed at the end point. For verifying the relationship between input air pressure and the bending angle, the soft actuator is driven by varied air pressures and the real-time signals of the L-TENG sensor are recorded (discussed in Supplementary Fig. 8, Fig. 9, and Supplementary Note 2). The air pressure starts from 50 kPa and increased by 10 kPa each step during the experiment. The result shows that the deforming angle of the soft actuator increases with the incremental air pressure, resulting in more output peaks generated by L-TENG sensor in one single actuation motion (Fig. 4c). Under the relative fast bending speed, the bending motion can still be clearly recognized by the details of generated output peaks as shown in Supplementary Fig. 8 (also see Supplementary Movie 4). The nearly linear relationship between the bending degree and peak number with increasing air pressure in Fig. 4c proves the

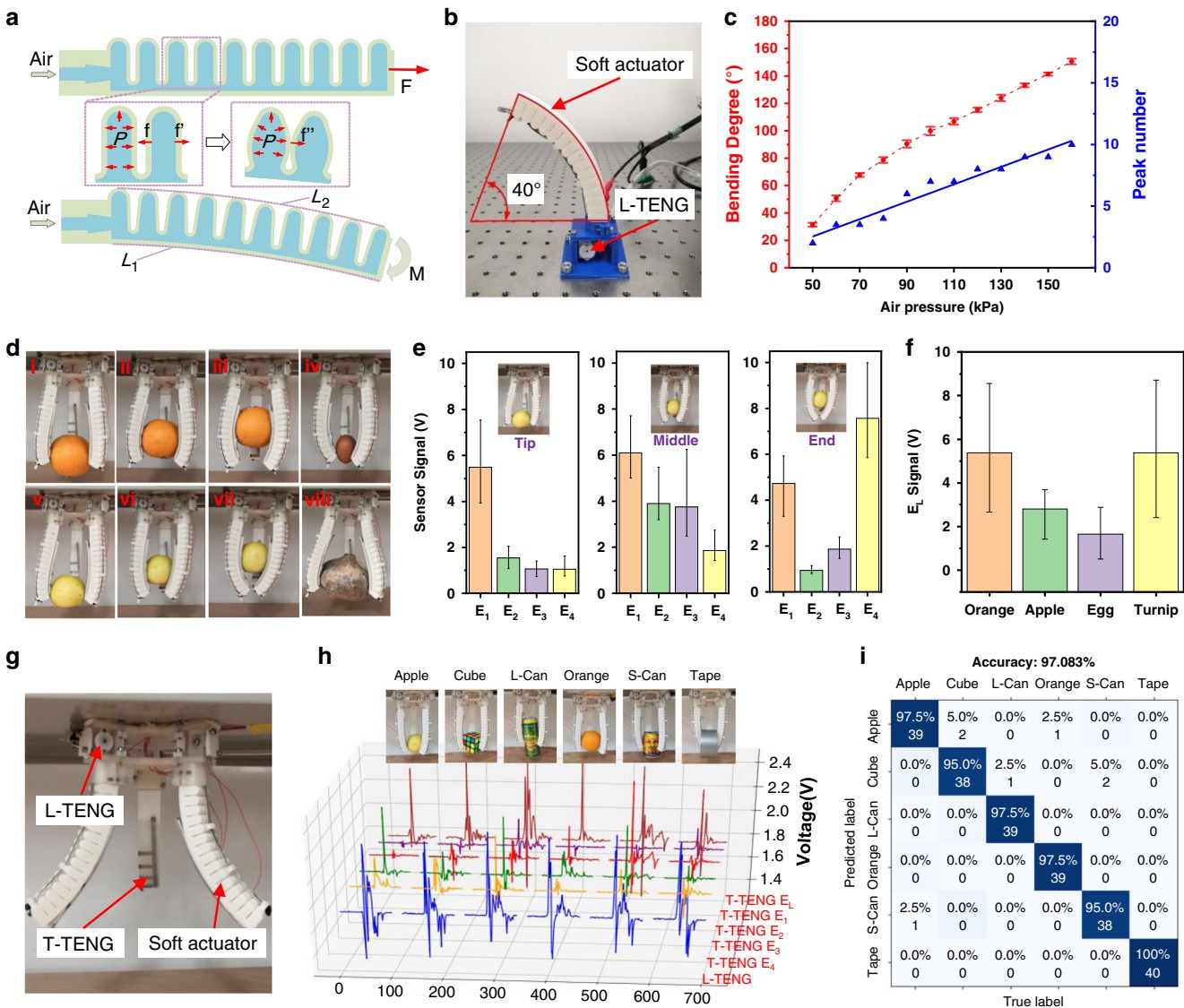

**Fig. 4 Verification of the soft actuator integrated with TENG sensors and the results of machine learning training. a** The operation mechanism of the soft actuator. **b** The configuration of the device during the characterization of length TENG (L-TENG) sensor. **c** The output peak numbers from L-TENG sensor versus the bending degrees of the soft actuator with increasing air pressure. **d** The diagrams of the gripper griping various objects with different contact positions to test T-TENG sensor. (i–iii) Gripping orange in different positions, (iv) gripping egg, (v–vii) gripping apple in different positions, and (viii) gripping turnip. **e** The output signals of tactile TENG (T-TENG) sensor for gripping an apple in various contact positions. **f** The output of long electrode ($E_L$) when gripping various objects with different curvatures. **g** The configuration of the gripper and sensors. **h** 3D plots of the robotic sensor outputs responding to different objects (apple, cube, long can (L-Can), orange, short can (S-Can), and tape). **i** The confusion map for machine learning outcome.

feasibility of measuring soft actuator's bending angle with L-TENG sensor's output signals. Moreover, the bending degree variations for soft actuator caused by L-TENG sensor can be ignored (discussed in Supplementary Figs. 10, 11, and Supplementary Note 3).

As mentioned above, the T-TENG sensor capable of sensing contact position has a good potential to be applied in the soft gripper. Here, a soft gripper composed of three soft actuators is shown in Fig. 4d. The T-TENG sensor patch is attached onto the inner surface of gripper's finger and contact with objects' surface directly during gripping. Here, considering the symmetrical structure of the soft gripper, only one finger is attached with the T-TENG sensor to reduce the sensory information complexity without losing important information. This T-TENG sensor-integrated gripper is successfully demonstrated to grasp objects of various dimensions including orange, green apple, egg, and turnip under the same input air pressure (160 kPa), as shown in Fig. 4d.

The output of the corresponding electrodes ($E_1$, $E_2$, $E_3$, and $E_4$) for the same object also varies due to different contact positions along the length direction of T-TENG sensor patch (Fig. 4e and Supplementary Fig. 12). When the apple is gripped at the tip position of the patch, the sensor signal in $E_1$ is higher than the others. Although when the apple is gripped at the middle position of the soft actuator, the output of $E_2$ and $E_3$ increases compared with the tip-contact situation, with the highest output still in $E_1$ and the lowest output in $E_4$. The reason is that the outputs of these four distributed short electrodes are inversely proportional to the distance between the contact position and the corresponding electrode. The closer the distance, the more charge will be induced in the electrode, thus generating a higher output. As for gripping the apple at a higher position, $E_4$ is the nearest electrode to the contact position and the tips of actuator fingers start to contact each other, in which case $E_1$ and $E_4$ have a higher output. The output ratio of these four short electrodes can then be

calculated as mentioned in Fig. 2d to better reflect the contact position of the soft actuator without environmental influences (Supplementary Fig. 12). In addition, the T-TENG sensor can also distinguish the contact and separation motions of the gripper by positive peaks and negative peaks respectively (Supplementary Fig. 12). By comparing the signals generated in $E_L$ when gripping objects with different dimensions in the tip-gripping mode, the difference in contact area can also be identified as shown in Fig. 4f, where a larger curvature can generate a higher output in $E_L$ due to a larger contact area. Therefore, this T-TENG sensor patch has a great potential to be integrated with soft grippers to help perceive the contact position, contact area, as well as distinguish the contact and separation motions when gripping objects. Signals generated by the soft gripper-embedded sensors vary with gripped objects' scale and contact position, verifying the feasibility of TENG-based sensor for soft-robotic applications.

**Data processing via ML technology.** The above work demonstrates that the proposed T-TENG and L-TENG sensors are capable of perceiving the bending degree, gripping position, and contact area as the sensing units for soft grippers. ML technology is an effective approach for dealing with classification problems[42,43] with complicated input signals and extracting features from data set automatically with certain algorithms (e.g., principal component analysis (PCA), t-distributed stochastic neighbor embedding, etc.) for further gripped objects recognition. Among the ML approaches, support vector machine (SVM) is a quite effective supervised learning model used for classification and has been proposed to be applied in analyzing triboelectric output signals with high performance[44]. Therefore, a customized SVM-based analytic platform is developed and then used in our TENG sensor-integrated smart soft gripper. The T-TENG sensor is attached directly onto the soft actuator surface without a PET substrate to ensure its stretchability. The detailed fabrication method and results are shown in Supplementary Fig. 13. Analog voltage signals generated in L-TENG and T-TENG sensor are first collected and processed by the hardware circuit consisting of an ADC and a micro control unit (MCU). Subsequently, the signals of both the T-TENG and L-TENG sensor are tested and recorded with corresponding object labels during the data collection process.

Because of the symmetrical structure of this soft gripper, a pair of single T-TENG sensor and L-TENG sensor are arranged on two different soft actuators as shown in Fig. 4g to reduce the information complexity and keep the important sensory information for classification. The detailed collected 6-channel (1 for L-TENG sensor and 5 for T-TENG sensor) signals for diverse objects are visualized in Fig. 4h and the data length for each channel is 100. In this process, the data of six objects with different shapes including apple, cube, long can (L-Can), orange, short can (S-Can), and tape is collected accordingly by repeating gripping and releasing motions. Here we directly use the raw voltage data in time domain of 6 channel as the feature of samples, so there are $100 \times 6 = 600$ features for each sample and each feature represents one data point at time series during grasping motion, which means the data includes the information of the contact force, speed, sequences, contact positions, latency, and the contact durations, etc. Those multi-dimensional features, rather than the individual grasping position, will then define the identity of the grasped object. Each object is gripped for 200 times to ensure the reliability of the data set in the data collection process. The 200 samples of each category are randomly split into two groups at the ratio of 8:2 (training: 160 samples, testing: 40 samples). Following the feature extraction process by PCA, the multi-class SVM classifier extended from classical two-class SVM classifier through the one-against-all principle is used to classify

these gripped objects based on the hundreds of data points extracted from the 600 data points of 6 channels for each sample. The verification results indicate the trained model has a high positive predictive value and a true positive rate for object recognition, and as shown in Fig. 4i, total recognition accuracy reaches 97.1%.

However, for a gripper with six sensing channels, there are only two-dimensional information regarding the contacted surface can be collected for gripped objects, through single T-TENG sensor and L-TENG sensor. Hence, this is more suitable to be used for symmetrical objects and the recognition ability is limited for more objects with different geometries. To enhance the performance, all of three fingers are then equipped with T-TENG sensors and L-TENG sensors, which can provide 15 channels of sensor output as shown in Fig. 5a. In Fig. 5b, 16 objects with different dimensions and curvatures are selected and tested in the ML process. Each object is gripped for 100 times to ensure the reliability of the data set. The 100 samples of each data set are randomly split into two groups at the ratio of 8:2 (training: 80 samples, testing: 20 samples). Figure 5c and Supplementary Fig. 14 present the outputs of sensors generated from the gripping of different objects. Supplementary Fig. 15 and Supplementary Note 4 illustrate the finger contact position for explaining the feasibility of identifying more objects. The data length for each channel is increased to 200 to capture more information during one grasping motion, so there are $200 \times 15 = 3000$ features for each sample. Although the types of objects increase, the confusion map in Fig. 5d shows that the trained model has a higher recognition accuracy of 98.1% compared to the previous six-channel model owing to the activation of more sensors when the penalty parameter C of the SVM is $1 \times 10^{-2}$ and the dimensionality of the data feature is reduced to 200 by PCA. The detailed parameter optimization of PCA and SVM can be found in Supplementary Table 2. As further verification, the data of similar objects (apple, orange, cube and small box, L-Can, and S-Can) are extracted from the data set of a 6-channel mode and compared with the corresponding data of 15-channel mode. After the training of the model based on the corresponding data (100 samples for each object) from these two modes, the comparison result is shown in Supplementary Fig. 16. It is obvious that the trained model with 15-channel has a higher prediction accuracy of 98.0% compared to that of the 6-channel model (94.0%), indicating that more sensor channels or useful data features play an important role in improving the recognition performance of the system. However, the increasing of sensor channels also brings higher cost and information complexity, which is a trade-off needed to be considered for practical applications. Then, based on the 15-channel model, the recognition results in Supplementary Table 3, Table 4, and Supplementary Note 5 validate the stability of our triboelectric sensory system under changing temperature and long-term use.

In terms of smart farms or factories, the sorting machines are frequently engaging with the objects which have similar shapes. To this concern, the data of sphere-shaped objects with various curvatures (baseball, tennis ball, apple, orange, and tangerine) is then selected for further discussion. Similarly, for understanding the influence of sensing channels on prediction accuracy, there are three modes applied for training, including pure L-TENG sensors of 3 fingers (3 channels), pure T-TENG sensors of 3 fingers (12 channels), as well as all of L-TENG and T-TENG sensors of 3 fingers (15 channels). According to the confusion maps in Fig. 5e–g, the accuracy of the ML model trained by 15 channels has the best accuracy of 95.0%, which is higher than that of the model with pure L-TENG sensors (77.0%) or T-TENG sensors (92.0%). It is obvious that the more sensor channels, the more useful sensory information can be collected and, thus, the

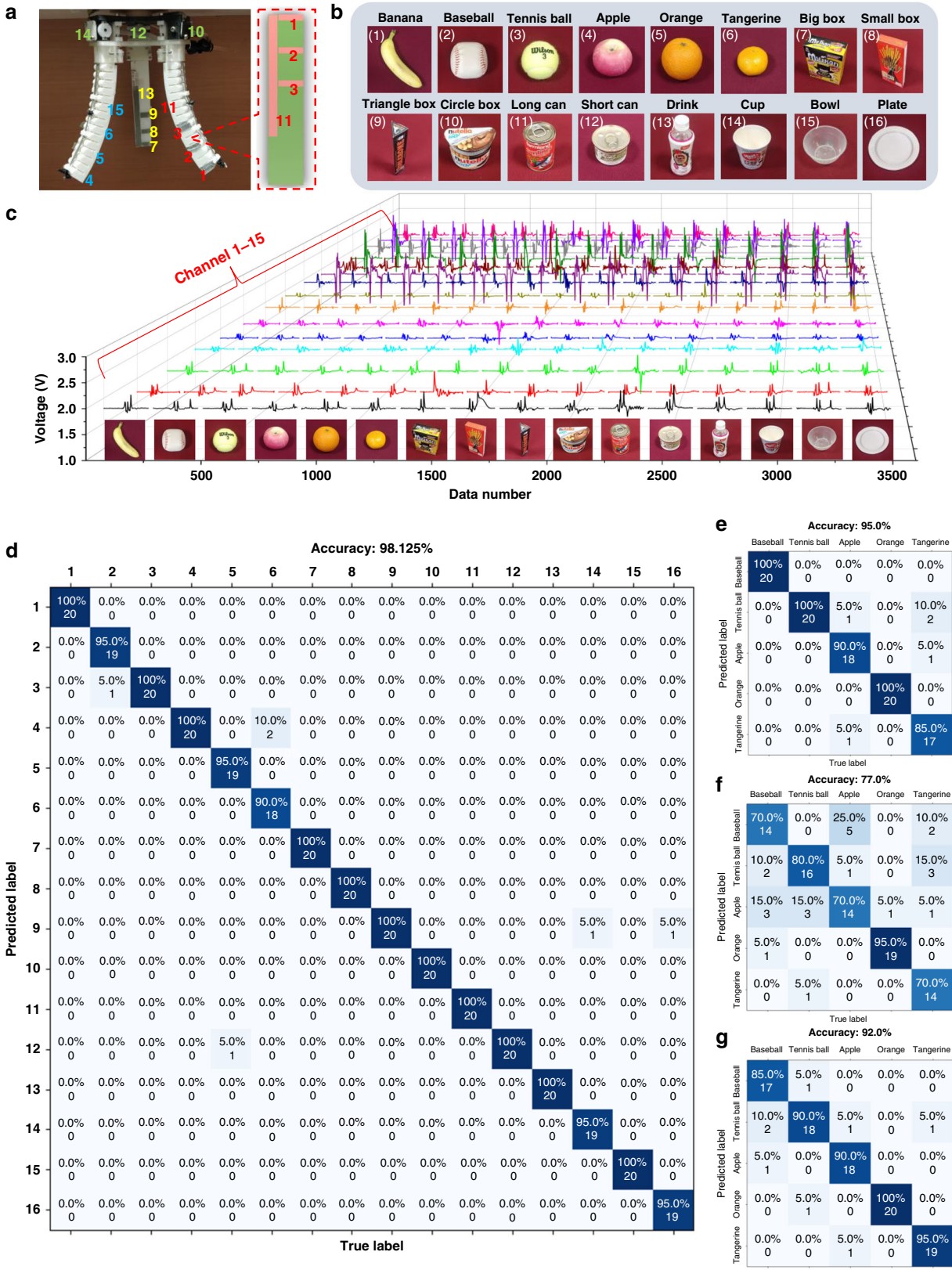

**Fig. 5 Enhanced object recognition system via machine learning technology. a** The channel arrangement of integrated sensors for collecting data. **b** The 16 objects to be gripped and recognized. **c** 3D plots of the robotic sensor outputs corresponding to different objects. **d** The confusion map for machine learning outcome of 16 objects with 15 channels. The comparison of confusion maps for machine learning outcomes of sphere objects with **e** all of 15 channels on both the length TENG (L-TENG) and tactile TENG (T-TENG) sensors, **f** pure L-TENG sensors (3 channels), and **g** pure T-TENG sensors (12 channels).

generated model is more powerful. For the model trained from pure L-TENG sensors, which mainly collect information of finger bending angle during grasping, the similar curvatures of baseball, tennis ball, apple, and tangerine may confuse the model. Hence, the result shows relative low accuracy. Besides, the three-channel information collected by pure L-TENG sensors also limits its recognition capability. Meanwhile, for T-TENG sensors that have 12 channels, the collected sensory information contains not only the gripping position but also the contact area. More useful features can be extracted and used to distinguish these similar objects, to achieve higher accuracy. The trained result in Supplementary Fig. 17 also verifies that the recognition accuracy is optimized with all of the 15 channels (T-TENG sensor + L-TENG sensor) and a relative higher accuracy is achieved by pure T-TENG sensors compared with pure L-TENG sensors.

**Digital twin demonstration with soft-robotic sensory gripper system.** To show the potential of using our soft-robotic sensory gripper system in future digital twin applications, a digital twin-based unmanned warehouse system is conceptualized, which can realize the automatic sorting and the real-time monitoring in a no-camera environment, where the recognition results in the real space can be real-time projected to a virtual space. Hence, this kind of soft-robotic sensory gripper system becomes an essential capability of implementing the mentioned applications[45]. The process flow of establishing and using the digital twin-based unmanned warehouse system is depicted in Fig. 6a. After trained with the SVM ML algorithm, the real-time object recognition system is achieved as shown in Fig. 6b (also see Supplementary Movie 5), where the screen displays both the real-time signals and

the corresponding predicted object's picture. Then, connect the real-time recognition system with a virtual space, the digital twin virtual projection application is also successfully implemented as shown in Fig. 6c (also see Supplementary Movie 6). When the system is running, the objects are randomly arranged to be gripped by the soft gripper, then the gripped objects will be recognized by the trained SVM model according to the input signals collected by L-TENG and T-TENG sensors. Both the signal waveform and the corresponding recognized result for each gripped object are shown on the top-left corner of the laptop screen, as shown in Fig. 6c. Then, the virtual objects deposited on the ground are gripped and placed in the corresponding boxes as the same objects gripped by the soft gripper and recognized by the prediction system in the real space.

## Discussion

In summary, a soft-robotic sensory gripper system integrated with two TENG sensors is proposed and investigated in this paper. Besides the contact-separation detection, the T-TENG sensor with patterned electrodes is able to sense the sliding, contact position, as well as the contact area of external stimuli via the triboelectric output of grating electrodes and $E_L$ electrode. Second, the L-TENG sensor offers the scalable measurement of bending motion and the integration of disc spring shows good recovery property, endowing the good sensing repeatability. By adopting the readout strategy of peak counting and output ratio, the influences of environmental fluctuation on the amplitude of signals can be effectively minimized, such as humidity, temperature, etc. The real-time signal processing ability of the sensory system also indicates the potential of realizing the

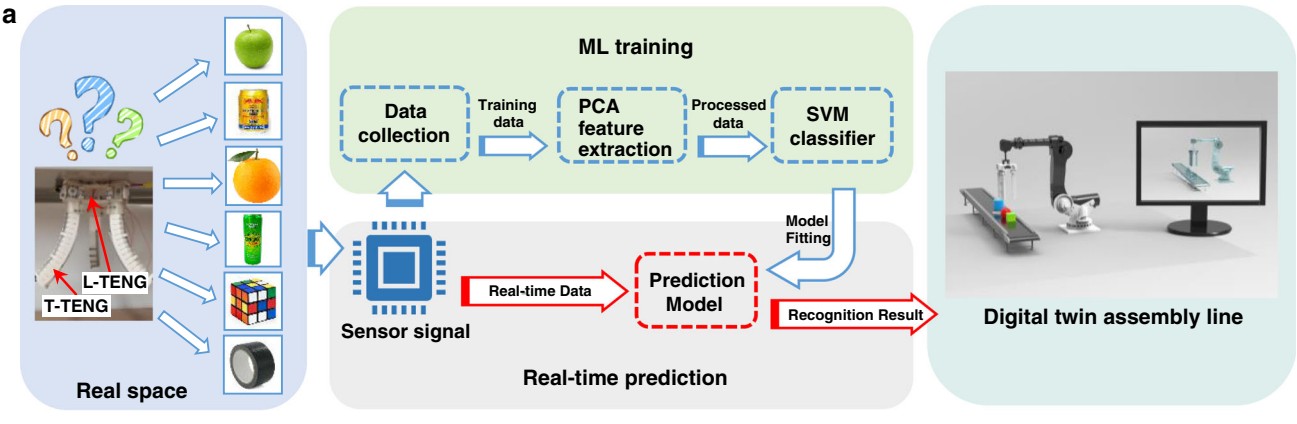

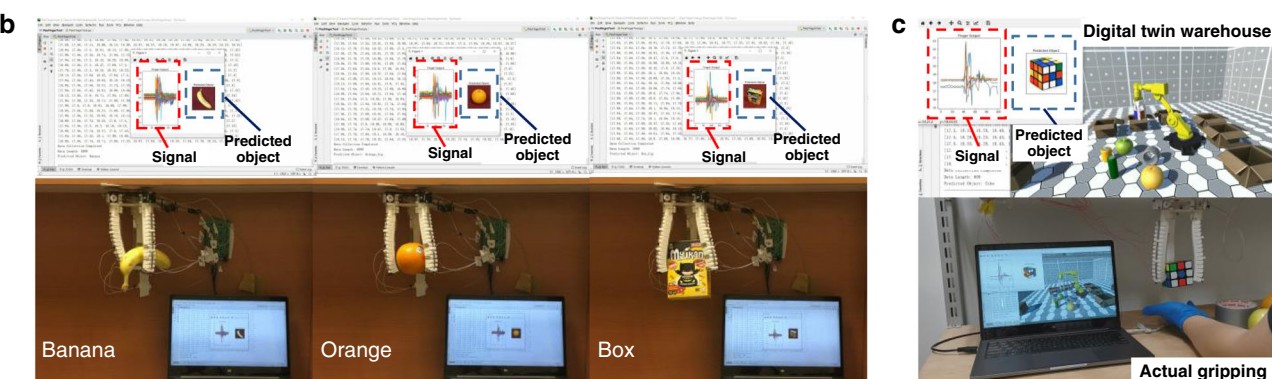

**Fig. 6 The length TENG (L-TENG) and tactile TENG (T-TENG) sensors integrated digital twin unmanned warehouse system with machine learning (ML) technology. a** The process flows from sensory information collection to ML training and real-time prediction in digital twin system, e.g., digital twin assembly line. The principal component analysis (PCA) is implemented to extract the data feature for ML training based on support vector machine (SVM). **b** Real-time gripped object prediction system. **c** The system interface integrated with object recognition and its digital twin warehouse application.

human–machine interactions. By leveraging the constant-curvature theory, we can estimate the bending degree of the pneumatic actuator when gripping the objects and, hence, to achieve the preliminary recognition. With the aid of the ML to deal with multi-parameter inputs, for fingers attached with single T-TENG and L-TENG sensor, the smart gripper successfully perceives and recognizes various objects with 97.1% accuracy, and can be further enhanced to 98.1% by increasing sensor channels from 6 to 15. Based on the improved intelligence of the soft gripper, the digital twin model simulates the robotic manipulation and the real-time object recognition in a duplicate virtual environment, and can be further applied to an assembly line for production control management in next-generation smart factories and shop floor management at unmanned warehouses. Besides, this TENG sensor-enabled soft-robotic gripper also shows promising application in other robots such as clawing robot, robotic arm, human-like hand, etc., for more effective and seamless human–machine interactions.

## Methods

**Fabrication of the TENG-based sensors**. The conductive textile (Ni-fabric) tape was cut into 5 mm × 25 mm and 5 mm × 120 mm to form the short and long electrodes before the electrodes were arranged on the PET film substrate (0.1 mm × 23 mm × 108 mm) by its own adhesive (acrylic pressure sensitive adhesive). Then, the substrate with the electrodes was fixed on the bottom of the mold with a 25 mm × 110 mm groove by double-sided tape as shown in Supplementary Fig. 1. After dispensing required amounts of Parts A and B of the EcoFlex 00-30 (Smooth-On), the mixed solution was poured to fill the groove on the mold as the negative triboelectric layer. The T-TENG sensor can peel off from the mold followed by a 60 min baking at 50 °C for curing. The main structure of L-TENG sensor was fabricated by a commercial 3D printer (4max pro, Anycubic). The positively triboelectric material (Ni-fabric) was coated on the gear by its own adhesive.

**Fabrication of the soft gripper**. The soft gripper with a bellows-structured actuator was designed by software (Solidworks 2016, Dassault Systèmes). To ensure elastic property and output force, TPU filament (NinjaFlex, NinjaTek) with a hardness of shore 85 A was used to fabricate the soft actuator. The filament was extruded by Titan Extruder and the printing temperature is 210 °C with corresponding low printing speed 30 m ms$^{-1}$. The flow filament diameter was decreased to 1.50 mm to enhance the amount of the extruded material and the layer thickness was changed to 0.15 mm, to guarantee the printing precision.

**Experiment measurement and characterization**. The signal outputs in the characterization of the T-TENG sensor were measured by an oscilloscope (DSO-X3034A, Agilent) using a high impedance probe of 100 MΩ. The open-circuit voltages were conducted by an electrometer (Model 6514, Keithley) and the signals were displayed and recorded by the oscilloscope. Analog voltage signals generated in L-TENG and T-TENG sensor were collected and processed by the hardware circuit consisting of an ADC and a MCU (Arduino Mega 2560). In the experiment concerned with the soft actuator, the air pressure was controlled by a reducing valve connected to a digital air pressure sensor (ISE30A, SMC) and the motion of the soft actuator was controlled by the solenoid valve.

## Data availability

The data that support the findings of this study are available from the corresponding authors upon reasonable request.

## Code availability

The codes that support the findings of this study are available from the corresponding authors upon reasonable request.

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

## Acknowledgements

This work was supported by the National Key Research and Development Program of China (Grant numbers 2019YFB1310901 and 2019YFB2004800), the National Natural Science Foundation of China (Grant numbers 61973207, 61673287, and 51705098), the Equipment Pre-Research Foundation for "the 13th Five-Year Plan" (Grant number 61403120306), and the HIFES Seed Funding-2017-01 (Grant number R-263-501-012-133) at the National University of Singapore.

## Author contributions

T.J. and L.L. designed and fabricated the soft actuator and soft-robotic gripper, designed and conducted the experiments, analyzed data, and wrote the manuscript. Z.S. and C.L. designed the control circuit, trained the machine learning model, and developed the digital twin system. T.J. and C.L. designed the triboelectric sensors. Q.Z., Z.Z., and G.Y. developed the experimental programs and edited the manuscript. M.Z. and T.C. designed and characterized T-TENG and L-TENG. Y.T., X.H., and C.L. directed the study and edited the manuscript.

## Competing interests

T.J., Z.S., L.L., and C.L. are inventors on patent application (pending, 10202003841W) submitted by National University of Singapore, which covers soft gripper perception using triboelectric sensors for digital twin.
