## [Peer Review File · Nature Communications]

Reviewers' comments:

Reviewer #1 (Remarks to the Author):

Disclaimer: I am not an author of any papers I mention in the review.

Paper Description

This paper describes two novel low-cost sensors for robotic perception, based on triboelectric nanogenerators (TENGs). In the introduction, the authors describe the need for smarter, low cost and high-information sensors for factories and industrial digital twins. They mention some disadvantages of modern sensors for robots with rigid fingers, and some of the sensors used in soft robotics. They then describe some methods of sensing in soft robotics (piezoelectricity, triboelectricity e.a.) and the existing use of TENGs in robotic sensing, combined with glove-based user-input devices. Their main thesis is that their developed sensors fill the gap in the sense that they are cheap to produce, convey large amounts of information, and contrary to exiting works they can detect contact, slippage, and contact type (point or area contact). They then describe their sensors: the length-TENG (L-TENG) sensor consists of a positive triboelectric component, which is a gear with four teeth. The negative component is an elastic strip of PTFE. As the strip is pulled, the gear's teeth momentarily touch the strip and result in the generation of a brief triboelectric voltage spike. The distance between spikes and their number, carry information related to the gear spinning angle and direction. A spring then ensures the smooth transition of the strip to the initial state. The tactile-TENG (T-TENG) sensor consists of five electrodes, one along the edge of a strip, and four across different points. Through triboelectric and electrostatic induction effects, the electrodes fire with different amplitudes when there is a contact. By comparing the electrode outputs, as well as the time difference between them, the sensor can detect contact and sliding motion along its length. The amplitude of the electrode along the edge provides info about the type of contact. The authors then demonstrate the usefulness of the sensors in a series of experiments. They control the bending angle and direction of a robotic finger with a glove that houses the two sensors. They then present a custom pneumatic soft robotic finger that can house the T-TENG sensor, calibrate it with and test the finger-sensor integration and the sensor output profiles by grasping a number of objects (apple, orange egg and turnip). They also show how the sensor signals extracted from grasping various objects (cube, cylinders, spheres and ring) can be used in tactile object recognition through SVM learning, achieving high recognition accuracy. Finally, they employ the SVM algorithm in a digital twin factory scenario, where a real soft gripper equipped with the T-TENG sensor recognizes various objects, and the digital twin robot sorts the object in different objects.

Paper strengths

Overall, the paper is well written. The introduction is well formed, and the authors incrementally build their arguments towards the novelty of their sensors and potential uses, and explain the merits compared to the existing literature. I believe that both the L-TENG and T-TENG sensors show very innovative designs, and the authors do a remarkable job in explaining how the output voltage signals are generated, and how these voltage profiles can be used to express robotic joint outputs and to detect slippage and contact detection. The experiments presented are simple and "to the point", clearly explaining their purpose and proving the arguments. The HMI control experiment provides a fundamental mapping of the L-TENG sensor output to a robotic joint position. I feel that the experiment would be more representative of robotic HMI control if the authors could show the continuous movement of the L-TENG on the HMI and the corresponding motion of the robotic joint, instead of 30-degree intervals, and I strongly encourage them to do so. The soft finger experiment helped showcase how the voltage profiles of the T-TENG sensor are used in object grasping. The object recognition experiment achieves very high accuracy, and I would like to see the results of it extended to new objects of varying curvatures (the YCB dataset would be a good test case [1]).

Paper shortcomings

The main criticism I have to offer is that the paper feels crammed. The authors try to present and solve a large range of problems related to joint position sensor design, tactile sensor design, robot teleoperation with HMIs, soft finger design, soft finger grasping, object recognition and learning, and digital twin applications. This is reflected to both the introduction section, where the authors list too many potential applications to solve, as well as the experimental methodology (i.e. lots of small-scale experiments to test a broad variety of issues). I believe that the digital twin experiment, albeit successful, was a bit unnecessary as it merely showed a grasping simulation executed with the SVM results as input. I would suggest that the paper could be broken into two papers: one sensor-oriented, that would describe analytically the sensors' principle of operation, the voltage profile generation, and the HMI control with more elaborate control methods, and one application oriented, where the authors could pick up a potential application (learning, digital twin or other) and demonstrate the usefulness of the sensors through thorough experiments and comparison with existing solutions of their chosen problems (for example, how does their T-TENG sensor perform compared to tactile sensors and skins in an object recognition experiment?).

Minor issues

The authors should check the spelling of some words throughout the manuscript and title.

Comments on the supplementary material

The supplementary material and schematics were useful to understand the soft finger operation. The video of the digital twin execution is simple and indicative. The video of the HMI is also clear, however it left me confused over the use of the T-TENG sensor. The authors claim in the paper that the T-TENG sensor is used to change the bending direction, however in the video it looks as if the T-TENG sensor does not affect the motion direction (the bending direction just seems to copy the motion of the glove finger). The authors should clarify a bit more how the T-TENG sensor is used on the HMI control experiment, and show it with some schematics or images.

References

[1] Calli, B., Singh, A., Bruce, J., Walsman, A., Konolige, K., Srinivasa, S., Abbeel, P. and Dollar, A.M., 2017. Yale-CMU-Berkeley dataset for robotic manipulation research. *The International Journal of Robotics Research*, 36(3), pp.261-268.

Reviewer #2 (Remarks to the Author):

The manuscript entitled "Development of Low-cost Human-Machine Interface Using Triboelectric Nanogenerator Sensors for Future Digital Twin" reported a self-powered intelligent warehouse system integrated with triboelectric nanogenerator (TENG) technology. Due to the synergistic effect of a length TENG (L-TENG) and a tactile-TENG (T-TENG), this smart system trained by machine algorithm exhibited an impressive capability of object identification with decent accuracy. Generally, this work is well illustrated with clearly organized figures and strong logics. The only concern is that this manuscript is mainly a fancy engineering technology demonstration, where TENG technology was used as a sensing interface to achieve artificial intelligence/machine learning. This application itself is not quite new, and there is almost no new fundamental understanding, new mechanism illustration, materials innovation, or novel discovery. The scientific value of this work is not significant. Again, it is a nice piece of engineering demonstration. It will be very helpful if the authors could elaborate more about technology advancements achieved in this work. Other minor concerns listed below should also be addressed:

1. Page 3, line 63-65, in the introduction section, the authors claimed that resistive sensors suffer from temperature instability that prevents their practicality. However, temperature also impacts TENG output

according to others' report (Advanced Engineering Materials 19.12 (2017): 1700275.). In order to show the superiority of TENG over other technologies, the authors are suggested show the accuracy of object identification under different temperature.

2. Page 4, line 88-90, the statement that soft materials of TENG and PENG has the same Young's modulus level as silicone and TPU is not correct. Piezoelectric soft polymer such as PVDF has Young's modulus up to 1 GPa, which is a few orders of magnitude higher than the modulus of soft silicone and TPU (1-10 MPa).

3. The measurements shown in Fig 3 are qualitative, such as tapping in the middle area between two electrodes will trigger smaller signals compared to direct tapping above the electrodes, and more contact area, lager output. This is all empirical. The contact speed and strain speed will all affect the output. Is there any fitting equation/curve could bridge position and output to predict the precise output at different position under different contact and strain speed? This might help further improve the accuracy.

4. How is the stability of the smart system? After cycles of utilization, is the accuracy still over 97%?

5. The authors should check typos before submission. The typos such as "nanaocomposites" in page 4, line 86 should be corrected.

Reviewer #3 (Remarks to the Author):

The major claims of the paper include the design of a flexible sensor called a L-TENG sensor to capture continuous motion of a soft gripper coupled with a T-TENG sensor for tactile sensing. The reviewer has listed several comments that will enhance the quality of the submission.

1. The work presented will be of interest to the audience of the journal and appears to be an extension of previous TENG advancements. The work described in the text along with the supplemental information indicate that the work can be reproduced by the authors.

2. The authors state, ". Compared to the previous solutions, the developed sensing system with patterned-electrode tactile TENG (T-TENG) sensor can detect sliding, contact position and gripping mode. " This statement would be better supported with quantitative comparison of the proposed new design with other solutions, optical etc. in a small table or figure.

3. The authors state, "With a layer of silicone rubber, a kind of flexible and stretchable negative triboelectric material, coated on the surface of the patch, triboelectric signals can be generated by the stimuli on the T-TENG sensor's surface and collected in these five electrodes," and couple this with a methods section. The process could be described in a more detailed manner and paired with the appropriate figure.

4. Since sensor and fabric application to the host influences detection and output signal, brief explanation of the method/type of adhesive used would be beneficial.

5. Figure 1 is difficult to read. In particular, the fonts are too small to read. Also, the relationship between elements in the figure are not clearly illustrated or described in the caption/text.

In summary, the work described is interesting and a meaningful advancement in TENG technology that can be applied to many applications.

Thank you for your letter and for the reviewers' comments concerning our manuscript entitled "Triboelectric Nanogenerator Sensors for Soft Robotics Aiming at Digital Twin Applications", which is modified from the original title of "Development of Low-cost Human-Machin-Interface Using Triboelectric Nanogenerator Sensors for Future Digital Twin", in order to reflect our main idea from the context precisely. Your comments are all valuable and very helpful for revising and improving our paper, as well as the important guiding significance to our research. We have studied comments carefully and have made the correction which we hope meet with approval. Briefly speaking, we have done extra experiment to collect new data in response to reviewers' comments. Thus, we have redesigned Fig. 1(a), 1(b) and 1(c), added Fig. 1(d). We have labeled the original Fig. 3 as new Fig. 2 and have merged Fig. 2(a) and 2(b) with Fig. 4(a), 4(c), 4(d) and 4(e) into new Fig. 3. The original Fig. 2(c), 2(d) and Fig. 4(b) have been moved into the Supplementary Materials and labeled as Fig. S4. We have labeled the original Fig. 5 as new Fig. 4, rearranged Fig. 4(d), 4(e), 4(f), and 4(i), moved original Fig. 5(g) to Fig. S12(d), and moved original Fig. 6(b) and 6(c) to Fig. 4(h) and 4(i). We have added new Fig. 5 for machine learning experiments, and the corresponding description into the result and discussion. We have also revised Fig. 6(a) and 6(b), and labeled the original Fig. 6(d) as Fig. 6(c). In Supplementary Materials, we have added new Fig. S1, S2 (with the description of Text S1), S14, and S15, added Table. S1, S2, S3, Text S2, S3, S4 and modified Fig. S13. The main corrections in the paper and the responses to the reviewer's comments are as following:

Reviewer 1

Comments 1: I feel that the experiment would be more representative of robotic HMI control if the authors could show the continuous movement of the L-TENG on the HMI and the corresponding motion of the robotic joint, instead of 30-degree intervals, and I strongly encourage them to do so.

Response:

We thank a lot for reviewer's valuable comment and suggestion. In terms of continuous motion sensing, there are several researches have demonstrated the continuous finger bending sensor, such as the grating patterned TENG sensor (Nano Energy, 2018, 54, 453–460), and many resistive based sensors (ACS Sens. 2016, 1, 817–825), etc. For our proposed L-TENG, the fundamental working principle is able to realize the continuous sensing. It has a resolution of 5mm. For the data glove presented in our paper, the length variation is limited by the finger deformation when the PTFE strip of L-TENG is pulled out by finger, and the number of contacts between the gear teeth and the strip is then limited for representing the displacement. Therefore, we realized the 30-degree interval. It is possible for higher accuracy as we mentioned in **Page 11** (i.e., increasing transmission ratio or the number of the gear

tooth, etc.). The L-TNEG with corresponding structure for improving precision is planned in our future work. To make it feasible, we can apply fabrication process with high precision, such as metallurgy or laser machining, in order to create fine pattern of teeth to enhance the resolution.

On the other hand, as our main novelty in this manuscript, we proposed tactile sensor (T-TENG) and length sensor (L-TENG) to form a perception system with machine learning technique for improving the intelligence of the soft gripper. **The peak counting and readout of output ratio are the data processing strategies which can eliminate the absolute amplitude variations of triboelectric output caused by environmental variations, and hence, to ensure the signal stability throughout the usage.** Moreover, by effectively utilizing the multi-dimensional data provided by two different sensors, the machine learning approach can provide equivalent precise measurement compared to continuous sensing regarding the object recognition, while avoiding the huge amount of data acquisition for continuous monitoring.

Comments 2: I would like to see the results of it extended to new objects of varying curvatures (the YCB dataset would be a good test case [1]).

Response: We appreciate the reviewer's suggestion on extending objects of varying curvatures. We agree that it would be better if more objects can be applied. In our previous work, only one T-TENG and one L-TENG sensor were applied to collect the data of 6 objects and we tested 200 times for machine learning training. It is feasible for our system to further enhance the recognition capability by using more sensors (each finger has one T-TENG sensor and one L-TENG sensor). Therefore, in response to the reviewer's advice, 16 types of objects with various shapes have been tested to construct the new data set, where we used 15 sensing channels (four channels of T-TENG and one channel of L-TENG on each of 3 soft actuators). Our result as depicted in Fig. 5 indicates that when the number of the objects was increased to 16, the recognition accuracy can still reach up to 98.1% for all the 16 types of objects with 15-channel signals. Besides, the model trained by data contracted from the previous data set showed that more applied samples can provide the better result (Page 19). Hence, we believe that our system can be used to recognize more objects of varying curvatures with the aid of introducing more sensor channels.

In the meantime, it is also worth to mention, in the practical application, such as fruit sorting in a specific company, the number of training objects for each job task is usually remain at few tens level. Hence, it will not be necessary for maintaining such high adaptability to all of the unnecessary objects in a certain scenario (i.e., unnecessary "pen" recognition for fruit sorting). In another word, for a small task including 5~10 objects, the minimalistic design with only one finger equipped with sensor is enough. For more complex tasks, we can then add extra sensors on other fingers to enable the scalability.

Actual change: The detailed content can be found in the Data Processing via Machine Learning Technology section from Page 16-20. Fig. 5, Fig. S14, Fig. S15, and Movie S5 were added to illustrate the concerned work.

Comments 3: The main criticism I have to offer is that the paper feels crammed. The authors try to present and solve a large range of problems related to joint position sensor design, tactile sensor design, robot teleoperation with HMIs, soft finger design, soft finger grasping, object recognition and learning, and digital twin applications. This is reflected to both the introduction section, where the authors list too many potential applications to solve, as well as the experimental methodology (i.e. lots of small-scale experiments to test a broad variety of issues).

Response: We thank a lot for reviewer's valuable comment and suggestion. To clarify our main idea in this manuscript, we propose a triboelectric nanogenerator (TENG) based smart sensory system for soft robotics, which consists of length TENG (L-TENG) and tactile TENG (T-TENG) sensors. Both of two types of sensors based on TENG technology were designed to solve the sensory problems of the soft gripper for digital twin applications. In Fig. 2 and 3, the presented results offer the basic characterizations of two sensors for understanding the working mechanisms and verifying the functionality with the real-time signal processing via a simple demonstration of glove-based sensing. Afterwards, we focused on the integration of the sensors into the soft gripper for enhancing the intelligence with the aid of machine learning algorithm, in order to realize the object recognition for digital twin application. In Fig. 5e-g, in response to reviewer's question about the necessity of combining several concepts, the results shown the importance of the fusion of two types of sensors by improving the recognition accuracy significantly, as each of two sensors offers the corresponding sensing dimension to increase the available features to be extracted for machine learning. In general, the digital twin is the potential application, and the soft finger is the main platform to perform digital twin. But it needs sensory system to achieve intelligent digital twin, which requires the joint position sensor (L-TENG) design and tactile sensor (T-TENG) design, while the robot teleoperation is just served as a preliminary verification of sensors' functionalities. In addition, we have rewritten our introduction and rearranged the sections of the manuscript for easier understanding where some content about the HMI has been delete.

Actual changes: The second and third paragraph of the introduction have been modified for expressing the necessity for developing soft robotic sensing and its background better. All the change has been highlighted from Page 3 to Page 5 in the manuscript and hope to dissipate reviewer's concern.

Comments 4: I believe that the digital twin experiment, albeit successful, was a bit unnecessary as it merely showed a grasping simulation executed with the SVM results as input. I would suggest that the paper could be broken into two papers: one sensor-oriented, that would describe analytically the sensors' principle of operation, the voltage profile generation, and the HMI control with more elaborate control methods, and one application oriented, where the authors could pick up a potential application (learning, digital twin or other) and demonstrate the usefulness of the sensors through thorough experiments and comparison with existing solutions of their chosen problems (for example, how does their T-TENG sensor perform compared to tactile sensors and skins in an object recognition experiment?).

Response: Thank you for reviewer's kind advice to our work. We apologize for the confusion of the main concept intended to present. As mentioned above, our main purpose is to develop a sensory system specifically for digital twin and capable of conducting object recognition with enough accuracy, which highly relies on the fusion of two types of sensors for soft robots to acquire enough sensible dimensions. All the sensor designs and technologies mentioned in the text are used for implementing an intelligent smart gripper system. In contrast, the presence of single type sensor will bring a severe decline of recognition accuracy and make the whole system unreliable (Fig. 5f and 5g). The glove-based demonstration is only served as a simple verification to show the feasibility of real-time signal processing for further researches on digital twin. Then, we tested the soft gripper integrated with designed sensors for robotic sensing. Finally, we showed the application of the soft gripper with the sensors for object recognition and concerned digital twin. This system has great potential for unmanned warehouse, farm or shop floor, while there is no light for camera sensing. Based on reviewer's comments, we have obtained a new data set by testing 16 types of objects with 15 channels (four for T-TENG and one for L-TENG on each soft actuator) and then trained the machine learning model by activating the data of L-TENG channels (3 channels), T-TENG channels (12 channels), and all the 15 channels respectively. The results indicate that the model trained by 15-channel data shows the best accuracy (98.1%) than the others. Meanwhile, the model trained by T-TENG channels' data performs better than the L-TENG's, especially for the object with similar curvature. Besides, we contracted the data of the previous data set and found that the model trained by 100 samples performs lower accuracy than the previous model. Therefore, we can conclude that the more sensory information, the more accurate the model will be and the sensory data from the T-TENG seems more important for object recognition. However, both the T-TENG sensor and L-TENG sensor contribute to the overall accuracy of the system.

Based on reviewer's comment, we also made some comparisons about the existing sensors for soft robotics by TENG technology. Prof. Wang's group firstly presented a soft robot with the TENG tactile sensor for proximity and pressure detection (*Adv. Mater.*, 2018, 30, 1801114). They discussed the contact position for robotic clawer by comparing the signals in different electrodes. In reference (*Adv. Mater. Technol.*, 2019, 1900337, 1–7), the soft gripper was covered by the TENG tactile sensor with only one electrode. They presented the signals as the gripper gripped or released the objects. In addition, Chen, S. *et al.* presented a wire-driven soft gripper with three TENG tactile sensors (*Adv. Mater. Technol.*, 2020, 1901075, 1–10) to detect the contact and compared the signals after gripping different objects. These existing solutions mainly concentrate in the view of the contact process or status (i.e., proximity, contact, gripping, and release) by evaluating the signals of single or multiple sensors. These sensors work separately and thus it is hard to perceive the accurate position (only right above the electrodes) or contact surface.

Actual Change: The detailed content can be found in the Data Processing via Machine Learning Technology section from Page 16-20. Fig. 5, Fig. S14, Fig. S15, and Movie S5 were added to illustrate the concerned work.

Comments 5: The authors should check the spelling of some words throughout the manuscript and title.

Response: We apologize for these mistakes in our manuscript. We have checked the spelling again and marked the modification.

Actual changes: The concerned content has already been highlighted throughout the manuscript.

Comments 6: The supplementary material and schematics were useful to understand the soft finger operation. The video of the digital twin execution is simple and indicative. The video of the HMI is also clear, however it left me confused over the use of the T-TENG sensor. The authors claim in the paper that the T-TENG sensor is used to change the bending direction, however in the video it looks as if the T-TENG sensor does not affect the motion direction (the bending direction just seems to copy the motion of the glove finger). The authors should clarify a bit more how the T-TENG sensor is used on the HMI control experiment, and show it with some schematics or images.

Response: We are sorry that the T-TENG in our manuscript is not well-explained. For verifying the real-time signal processing system, we use a slice of T-TENG with single electrode to switch the motion direction of the robotic hand and it is mounted on the thumb position of the glove, which has been mentioned in Page 12. We have modified the Fig. 3f(□) to express the location of the T-TENG better. As depicted in Fig. 3f(□), the motion direction of robotic finger will be changed after the trigger caused by the tapping of the thumb and index finger. Besides, the position of the both sensors has been marked in the supplementary movie (Movie S3). We hope these modifications may dissipate review's confusion.

Actual changes: The concerned content can be found in Page 12, which has been highlighted. And the modified movie can be found in supplementary material (Movie S3).

Reviewer 2

Comments 1: This application itself is not quite new, and there is almost no new fundamental understanding, new mechanism illustration, materials innovation, or novel discovery. The scientific value of this work is not significant.

Response:

We thank a lot for reviewer's valuable comment. In this paper, we proposed a tactile sensor (T-TENG), a length sensor (L-TENG), and a perception system with machine learning technique for improving the intelligence of the soft gripper. To indicate the scientific value, there are several points which can be clarified.

1. We integrated the triboelectric sensor on pneumatic finger to realize self-powered sensing, which can reduce the power consumption for massive deployment, especially in wireless AIoT framework. There are only a few papers studied

triboelectric based sensing for pneumatic actuator or continuous motion sensing. But they do not possess the capability of multi-dimensional sensing for practical application, especially digital twin.

2. We firstly proposed the gear-based length sensor (L-TENG) to measure the bending degree of the soft gripper, which has the specially designed disc spring for recovery after air off for the soft actuator. The special gear-based structure of the L-TENG sensor has not been proposed before and can be used for continuously sensing of the displacement which is difficult to be achieved by common triboelectric based sensors. The resolution can also be further improved as we mentioned Page 11 by adjusting the transmission ratio or gear tooth number, etc.
3. T-TENG can not only detect the siding, contact position but also contact area with simple structure, which is useful to distinguish surface contact and point contact for soft gripper.
4. Refer to Fig. 2, we proposed the special design of T-TENG electrodes with the readout method of output ratio. Together with peak counting readout of L-TENG shown in Fig 3, we can solve the problem of output amplitude fluctuation caused by environmental noises.
5. We successfully utilized the constant curvature theory to explain the bending angle of pneumatic finger by the outputs from TENG sensor, and obtained the results shown in Fig. 4c. This approach verifies the feasibility of realizing primary recognition without advanced data processing.
6. For more complex tasks in practical application, we investigated the object recognition assisted by the machine learning technique. To achieve this task, we effectively combined L-TENG and T-TENG to create the enough dimensions of sensing data for increasing the recognition accuracy, as shown in Fig 5.
7. The proposed sensory system should have potential for unmanned warehouse, farm land and shop floor management, where there is no light or no camera, as shown in our integrated demonstration.
8. The object recognition together with sensory system also helps to build up a comprehensive database by the continuous accumulation of sensing data during million times of grasping of diverse objects in practical application. These collected massive data become a valuable guideline and evaluation about the performance of grabbing object by robots, such as the contact mapping which reflects the most frequent touched areas or less touched areas, as well as the slippery area. This information is crucial for those robotic arm engineers to improve the designs accordingly.

Comments 2: Page 3, line 63-65, in the introduction section, the authors claimed that resistive sensors suffer from temperature instability that prevents their practicality. However, temperature also impacts TENG output according to others' report. In order to show the superiority of TENG over other technologies, the authors are suggested show the accuracy of object identification under different temperature.

Response: Thank you for reviewer's kind advice. We agree with the reviewer's

opinion that the working temperature may affect the object identification. However, it usually happens at very high or low temperature. As mentioned in some literature data (Nano Energy, 2014, 4, 453–460; Appl. Phys. Lett., 2015, 106, 013224), the triboelectric output will experience the significant fluctuation after reach above 320K or below 200K. Hence, the triboelectric based sensor can provide stable functions under the practical scenarios, where the temperature seems to remain in a constant value (298K) and the fluctuation of temperature is quite small. As the fundamental principle of machine learning algorithm is to identify the signal patterns and extract the distinct features among various objects. Hence, for a specific object, the system can recognize it successfully for each cycle if the output of contacting can maintain the consistency of the signal patterns, which means the high accuracy. As reviewer’s kind advice, we have tested the signals for some of the objects in diverse temperatures (298K and 313K). As depicted in Table 1, our results indicate that within normal ambient temperature, our TENG sensory system is not very sensitive to the temperature. In addition, we can also improve the accuracy of the model by getting more data under different temperatures for training to enhance the versatility of the model when the environment has large temperature variation.

Actual Change: Table 1 in reply can also be found in the Supplementary Materials (Table S2)

Table 1 The real-time object recognition results under different temperatures

Temperature	Small box	Orange	Apple	Long Can	Short Can	Accuracy
298K	20/20	19/20	19/20	20/20	19/20	97/100 (97%)
313K	20/20	17/20	18/20	19/20	20/20	95/100 (95%)

Note: “19/20” means that 19 times correct recognition in 20 times gripping tests.

Comments 3: Page 4, line 88-90, the statement that soft materials of TENG and PENG has the same Young’s modulus level as silicone and TPU is not correct. Piezoelectric soft polymer such as PVDF has Young’s modulus up to 1 GPa, which is a few orders of magnitude higher than the modulus of soft silicone and TPU (1-10 MPa).

Response: We apologize for the mistake in our manuscript. Sensors made of PVDF film can also show good flexibility if it is thin enough, although it seems hard to be stretched. Therefore, thin PVDF film can also be used in soft gripper if it has a stretching limitation layer to make the soft actuator bend. Thanks for reviewer’s advice and we have modified the corresponding expression.

Actual Change: The concerned content can be seen in Page 3.

Comments 4: The measurements shown in Fig 3 are qualitative, such as tapping in the middle area between two electrodes will trigger smaller signals compared to direct tapping above the electrodes, and more contact area, lager output. This is all empirical. The contact speed and strain speed will all affect the output. Is there any fitting equation/curve could bridge position and output to predict the precise output at different position under different contact and strain speed? This might help further improve the accuracy.

Response: We would like to express our most gratitude to the reviewer's kind advice. For distinguishing the signal caused by different contact and strain speed, we used the formula $Ratio = E_i / \sum_{i=1}^4 E_i$ as mentioned in Page 9, which presents the ratio of the output value of the single electrode to all of the four short electrodes.

Fig. R1 The simplified diagram of the contact process for electrostatic analysis.

The main working mechanism is listed as following. According to the electrostatic induction, we can define the electric potential as

$$U = k \frac{Q}{r} \quad (1)$$

where Q is the amount of charge, r is the distance to the point charge, and k is Coulomb's constant as shown in Fig. R1 below. So, if we assume that a charge of $+Q$ moves to the silicone rubber surface with a distance of h , the output voltage of two nearby electrodes can be expressed as

$$\begin{cases} V_{E_i} = k \frac{Q}{\sqrt{x^2 + h^2}} - k \frac{Q}{x} \\ V_{E_{i+1}} = k \frac{Q}{\sqrt{(l-x)^2 + h^2}} - k \frac{Q}{l-x} \end{cases} \quad (2)$$

where x represents the distance between the touch point to E_i and $l-x$ represents the distance between the touch point and E_{i+1} . Thus, the ratio can be derived as

$$\frac{V_{E_{i+1}}}{V_{E_i}} = \frac{\frac{1}{\sqrt{(l-x)^2 + h^2}} - \frac{1}{l-x}}{\frac{1}{\sqrt{x^2 + h^2}} - \frac{1}{x}} \quad (3)$$

So if h is large enough, we can obtain

$$\frac{V_{E_{i+1}}}{V_{E_i}} = \frac{x}{l-x} \quad (4)$$

Therefore, the ratio can be used to explain the method to get a more accurate position under virous speeds and strain speeds. Similarly, as the electrodes number are increased to four, we can use $Ratio = E_i / \sum_{i=1}^4 E_i$ to sense the position. As depicted in

Fig.2d, we analyzed the data after contacting right above the electrode and the middle

position of two electrodes. The result shows the reliability of the T-TENG to sense the contact position.

Actual Change: The concerned content can be seen in Page 9 and we have added the detailed explanation in Text S1 in Supplementary Materials.

Comments 4: How is the stability of the smart system? After cycles of utilization, is the accuracy still over 97%?

Response: We would like to express our most gratitude to reviewer's effort and patience in reviewing our manuscript. According to the comment, we have used the soft gripper integrated with both L-TENG and T-TENG sensors (16 channels) to grip various objects with 2000 times to show the stability of our sensory system. In order to reduce the proportion of necessary training and testing times in the entire stability test and make the result for certain utilization cycles more accurate, we choose to use less data: 80 samples for training for each object (total 5 objects) and 20 samples for testing for each object after cycles of utilization. As shown in Table 2 below, our results indicate that though the recognition accuracy has some inevitable fluctuation due to the environmental and man-made noise, the average accuracy remains high level and there isn't an obvious decrease after gripping for 2000 times. Therefore, we believe that our design can keep a high accuracy even though after cycles of utilization.

Actual changes: Table 2 in reply can also be found in Supplementary Materials (Table S3)

Table 2 The test results for verifying the stability of the smart system

	Gripping Counts								
	100	300	500	700	1000	1300	1500	1800	2000
Small Box	20/20	20/20	20/20	19/20	20/20	18/20	20/20	20/20	20/20
Orange	19/20	18/20	19/20	18/20	17/20	18/20	17/20	19/20	19/20
Apple	19/20	19/20	19/20	20/20	20/20	19/20	19/20	19/20	19/20
Long Can	20/20	20/20	18/20	19/20	18/20	20/20	20/20	20/20	19/20
Short Can	18/20	20/20	20/20	20/20	19/20	20/20	19/20	19/20	19/20
Accuracy	96/100 (96%)	97/100 (97%)	96/100 (96%)	96/100 (96%)	94/100 (94%)	95/100 (95%)	95/100 (95%)	97/100 (97%)	96/100 (96%)

Note: "19/20" means that 19 times correct recognition in 20 times gripping tests.

Comment 5: The authors should check typos before submission. The typos such as "nanaocomposites" in page 4, line 86 should be corrected.

Response: We apologize for these mistakes in our manuscript. We have checked the spelling again and marked the modification.

Actual changes: The concerned content has already been highlighted throughout the manuscript.

Reviewer 3

Comment 1: “Compared to the previous solutions, the developed sensing system with patterned-electrode tactile TENG (T-TENG) sensor can detect sliding, contact position and gripping mode.” This statement would be better supported with quantitative comparison of the proposed new design with other solutions, optical etc. in a small table or figure.

Response: Thanks for reviewer’s kind advice to our work and sorry for our mistakes. In this paper, we want to introduce the sensing methods for soft gripper which requires special design for sensors. Actually, we want to express that compared to the previous solutions based on TENG technology for robots, our device can detect sliding, contact position and gripping mode. The concerned sentence has been modified (Page 4). Meanwhile, the T-TENG sensor featuring simple structure and low cost has its advantages compared to other methods. As reviewer’s kind comment, some comparisons about the tactile sensor based on TENG technology and other methods (e.g., piezoelectricity) have been listed in Tab.1 below.

Actual changes: The comparison of the tactile sensors can be found in Table 3 below (also see Table S1 in Supplementary Materials). The concerned content has been modified in Page 4.

Table 3 Comparison of the tactile sensor based on TENG technology and other methods

Method	Position	Sliding	Contact Surface	Force	Electrode Number	Application
Triboelectric	Simple (Contact or separation)	No	No	No	One	Soft gripper ¹⁻³
Triboelectric	5×5 sensory array	Yes	No	No	Four	Robot control ⁴
Triboelectric	8×8 sensory array	Yes	Yes	No	Sixteen	Wearable device ⁵
Triboelectric	Simple (Contact or separation)	No	No	No	Two or three	Wearable device ^{6,7}
Capacitive, Triboelectric	4×4 sensory array	Yes	No	Normal	Eight	Humanoid Hand ⁸
Piezoresistive	No	No	Yes	No	One	Soft gripper ⁹
Piezoresistive	Yes	No	No	Yes	One	Soft gripper ¹⁰
Piezoresistive	Yes	No	Yes	No	Two or more	Soft gripper ¹¹
Piezoelectric	No	Yes	No	No	Two	Textile detection ¹²
Piezoelectric	No	No	No	Contact or not	One	Wearable device ¹³
Capacitive	Yes	No	No	No	Four	HMI ¹⁴
Capacitive	No	No	No	3-axis	Four	Humanoid Hand ¹⁵
Capacitive	Yes	No	No	Contact	Six	Wearable device ¹⁶

				or not		
Thermosensitive	No	No	No	Normal	One	Humanoid Hand ¹⁷
Optical	Yes	No	No	Normal		Sensor Mechanism ¹⁸

Comments 2: The authors state, "With a layer of silicone rubber, a kind of flexible and stretchable negative triboelectric material, coated on the surface of the patch, triboelectric signals can be generated by the stimuli on the T-TENG sensor's surface and collected in these five electrodes," and couple this with a methods section. The process could be described in a more detailed manner and paired with the appropriate figure.

Response: Thanks for reviewer's kind advice to our work. We have added a figure about the fabrication process in Supplementary Materials (Fig. S1), which may do some helps. Besides, the methods part has been added some detailed parameters about the fabrication of the T-TENG sensor. Hope to dissipate the reviewer's concern

Actual changes: These changes can be found in the Methods section (Page 22) and the latest supplementary materials (Fig. S1).

Comment 3: Since sensor and fabric application to the host influences detection and output signal, brief explanation of the method/type of adhesive used would be beneficial. **Response:** We appreciate the reviewer's suggestion on clarifying the method/type of adhesive used in our manuscript. At first, we used the PET film as the substrate to test the T-TENG's working mechanism. The conductive textile tape was pasted on the PET film by its own adhesive, a kind of acrylic pressure sensitive adhesive. Then, the EcoFlex 00-30 was used to cover and seal the Ni-fabric layer by a mold (see Fig.S1 in Supplementary Materials). Similarly, the same method has been used in the integration of T-TENG and the soft actuator as illustrated in Fig. S13 where the extra silicone rubber was used to reinforce the adhesion of the EcoFlex and TPU.

Actual Changes: The methods for fabrication T-TENG have been modified based on reviewer's advice Page 22. Besides, Fig. S1 has been added and the Fig. S13 has been modified for better explanation.

Comments 4: Figure 1 is difficult to read. In particular, the fonts are too small to read. Also, the relationship between elements in the figure are not clearly illustrated or described in the caption/text.

Response: We apologize for these mistakes in our manuscript. The concerned figure has been modified according to reviewer's advice. Thank you for reviewer's kind advice

Actual changes: These changes can be found in Fig. 1 of the latest manuscript.

Reference:

1. Chen, J., Chen, B., Han, K., Tang, W. & Wang, Z. L. A Triboelectric Nanogenerator as a Self-Powered Sensor for a Soft-Rigid Hybrid Actuator. Adv. Mater. Technol. 1900337, 1–7 (2019).

2. Lai, Y. et al. Actively Perceiving and Responsive Soft Robots Enabled by Self-Powered, Highly Extensible, and Highly Sensitive Triboelectric Proximity- and Pressure-Sensing Skins. *Adv. Mater.* 1801114, 1–12 (2018).
3. Chen, S., Pang, Y., Yuan, H., Tan, X. & Cao, C. Smart Soft Actuators and Grippers Enabled by Self-Powered Tribo-Skins. *Adv. Mater. Technol.* 1901075, 1–10 (2020).
4. Chen, T. et al. Triboelectric Self-Powered Wearable Flexible Patch as 3D Motion Control Interface for Robotic Manipulator. *ACS Nano* 12, 11561–11571 (2018).
5. Dong, K. et al. A Stretchable Yarn Embedded Triboelectric Nanogenerator as Electronic Skin for Biomechanical Energy Harvesting and Multifunctional Pressure Sensing. *Adv. Mater.* 30, 1–12 (2018).
6. Yang, P. et al. A Flexible, Stretchable and Shape-Adaptive Approach for Versatile Energy Conversion and Self-Powered Biomedical Monitoring. *Adv. Mater.* 27, 3817–3824 (2015).
7. Wang, H. et al. Self-Powered Inhomogeneous Strain Sensor Enabled Joint Motion and Three-Dimensional Muscle Sensing. *ACS Appl. Mater. Interfaces* 11, 34251–34257 (2019).
8. Zhang, C. et al. A stretchable dual-mode sensor array for multifunctional robotic electronic skin. *Nano Energy* 62, 164–170 (2019).
9. Homberg, B. S., Katzschmann, R. K., Dogar, M. R. & Rus, D. Haptic identification of objects using a modular soft robotic gripper. *IEEE Int. Conf. Intell. Robot. Syst.* 2015-Decem, 1698–1705 (2015).
10. Thuruthel, T. G., Shih, B., Laschi, C. & Tolley, M. T. Soft robot perception using embedded soft sensors and recurrent neural networks. *Sci. Robot.* 4, eaav1488 (2019).
11. Hughes, J. & Iida, F. Tactile Sensing Applied to the Universal Gripper Using Conductive Thermoplastic Elastomer. *5*, 512–526 (2018).
12. Liao, Z. et al. A tactile sensor translating texture and sliding motion information into electrical pulses. *Nanoscale* 7, 10801–10806 (2015).
13. Deng, W. et al. Cowpea-structured PVDF/ZnO nanofibers based flexible self-powered piezoelectric bending motion sensor towards remote control of gestures. *Nano Energy* 55, 516–525 (2019).
14. Peele, B. et al. Untethered Stretchable Displays for Tactile Interaction 1. *6*, 142–149 (2019).
15. Liang, G., Wang, Y., Mei, D., Xi, K. & Chen, Z. Flexible Capacitive Tactile Sensor Array with Truncated Pyramids as Dielectric Layer for Three-Axis Force Measurement. *J. Microelectromechanical Syst.* 24, 1510–1519 (2015).
16. Ai, Y. et al. All rGO-on-PVDF-nanofibers based self-powered electronic skins. *Nano Energy* 35, 121–127 (2017).
17. Li, G. & Zhu, R. A Multisensory Tactile System for Robotic Hands to Recognize Objects. *Adv. Mater. Technol.* 1900602, 1900602 (2019).
18. Massari, L. et al. A Machine-Learning-Based Approach to Solve Both Contact Location and Force in Soft Material Tactile Sensors. *Soft Robot.* 00, 1–12 (2019).

REVIEWER COMMENTS

Reviewer #1 (Remarks to the Author):

I would like to thank the authors for addressing my comments and the comments of other reviewers, and extensively describing the changes they made. Their explanatory responses and manuscript additions in my concerns for the length of the paper and necessity of the digital twin experiment made the manuscript richer and more streamlined.

The machine learning experiment added to section "Data Processing via Machine Learning Technology" offers additional insight over the sensor performance in gripping tasks. The results seem to validate the usefulness of the two TENG sensors in grasping tasks of simple objects, with confusion matrices that reflect good performance. The authors need to describe the data collection and training procedure of the SVM: a) The authors claim that they used the voltages of electrodes and applied PCA to extract features, however they should mention exactly what features they have used as input to SVM (such as voltage peaks, RMS value etc.), and technical details on the SVM (kernel used etc). b) The authors need also to describe exactly how the grasping motion is performed to gather data for the 100 grasps on each object, and the finger contact location on the object. If the grasping motion is performed the same way as shown in Movie S5, there is a good chance that the 100 grasps for each object are similar to one another. The 80:20 split over all objects is also a factor for the good performance, as the usage of all objects in the training process ensures that the SVM recognises data very similar to the ones that have been trained upon. I understand that the authors argue their system is suited for applications where the robot encounters the same object (such as fruit picking), and such a method of training is viable. I suggest the authors conduct an experiment where all data from 10-11 objects are used in training and all data from the rest 5-6 objects are used in testing. This will enable the research community understand the applicability of their sensors in more general object recognition, where generalisation to unknown objects is required.

Reviewer #2 (Remarks to the Author):

In the revised manuscript, the authors clearly addressed all the concerns I raised. It can be seen that the authors made a great effort to clarify all uncertainties and confusing points in the manuscript. It is an impressive engineering development of a TENG-based technology, while I still don't think the scientific impacts and novelty are very high. Nonetheless, the quality of work in this manuscript is high and can be recommended for publication on Nature Communications.

Reviewer #3 (Remarks to the Author):

The reviewer has reviewed the submission as well as the rebuttal to the original reviewer comments. The reviewer finds that the authors have effectively addressed the comments of the reviewers.

Dear Reviewers:

Thank you for your letter and for the reviewers' comments concerning our manuscript entitled "Triboelectric Nanogenerator Sensors for Soft Robotics Aiming at Digital Twin Applications". Your comments are all valuable and very helpful for revising and improving our paper, as well as the important guiding significance to our research. We have studied comments carefully and have made the correction which we hope meet with approval. Briefly speaking, we have done extra experiment to collect new data in response to reviewers' comments. In Supplementary Materials, we have added new Fig. S14, S17 (with the description of Text S5), added Table. S4, labeled the original Fig. S14 as new Fig. S15 and labeled the original Fig. S15 as new Fig. S16. The main corrections in the paper and the responses to the reviewer's comments are as following:

Comment 1: The authors claim that they used the voltages of electrodes and applied PCA to extract features, however they should mention exactly what features they have used as input to SVM (such as voltage peaks, RMS value ea.), and technical details on the SVM (kernel used etc).

Response:

We thank a lot for reviewer's valuable comment. For the input of the PCA¹, we used the raw data of 15 channels of each grasp which is shown in Fig. R1 (Different columns show the time-domain signals of 15 channels of different objects during one grasping motion). The data length for each channel is 200, so there are $200 * 15 = 3000$ features for each sample and each feature means one data point in the time domain during grasping. Then the PCA will extract features and reduce the dimensionality from 3000 to a few hundred, that means the final input to SVM² will be hundreds of data points of each sample which can best distinguish these objects. For better understanding, the enlarged voltage waveforms of each channel for grasping the banana are shown in Fig. R2, where the outputs of channel 1-9, 11, 13, 15 are from T-TENG sensors, and the outputs of 10, 12, 14 are from L-TENG sensors. All the data points of these channels will be directly used as input to the ML process. Due to the larger contact area when grasping the banana, channel 2, 3, 11 and 13 shows greater voltage value than other channels among T-TENG sensors, which can be further visualized as peak voltage maps as the reference to distinguish different objects by the human eye shown in Fig. R3. But the machine distinguishes these objects based on the hundreds of data points extracted from the 3000 data points of 15 channels for each sample, not just the voltage peaks, peak values or RMS values. The technical details of the SVM can be found in Table 1. We used the linear kernel and optimized the accuracy of the SVM according to commonly used parameters, and found that the optimal recognition accuracy of 98.125% can be achieved when the penalty parameter C is $1 * 10^{-2}$ and dimensionality of the data feature is reduced to 200 by PCA.

Actual Change: The concerned content has already been added and highlighted in Page 18, line 11-25 and Page 19, line 12-19. Fig. R1 and Table 1 in reply are provided in the Supplementary Materials (Fig. S14, Table S4).

Fig. R1. The input voltage signals of 15 channels for 16 different grasped objects.

Fig. R2. The input voltage signals of 15 channels for grasping the banana.

Table 1 Parameter optimization of SVM and PCA

Classification accuracy		Penalty parameter C			
		1×10^{-2}	1×10^{-1}	1	10
Linear kernel	PCs = 100	96.25%	95.938%	95.938%	95.938%
	PCs = 120	96.563%	96.563%	96.563%	96.563%
	PCs = 150	97.5%	97.5%	97.5%	97.5%
	PCs = 200	98.125%	97.813%	97.813%	97.813%
	PCs = 250	97.5%	97.5%	97.5%	97.5%

Comment 2: The authors need also to describe exactly how the grasping motion is performed to gather data for the 100 grasps on each object, and the finger contact location on the object. If the grasping motion is performed the same way as shown in Movie S5, there is a good chance that the 100 grasps for each object are similar to one another.

Response:

Thank you for reviewer's kind advice. In terms of the primary demonstration of showing the feasibility for object recognition in warehouse or factory where the robotic gripper usually grasps products with the same position and angle, we try to maintain a relatively fixed position for 100 grasps for each object during the data collection process to achieve a better prediction result with less samples. The peak voltage maps of the T-TENG sensors for different objects shown in Fig. R3 provide a reference for the finger contact location on the object. The peak voltages of the 12 T-TENG sensors for each sample are normalized between zero and one, and a darker color means larger contact area and contact force at this location during grasping. The grasping pressure differences among the 12 T-TENG sensors between objects are also clearly illustrated in this figure. Though the fabrication errors existing in our homemade sensors and pneumatic fingers may result in the asymmetrical distribution of the peak voltages in the T-TENG sensors for symmetric objects, this will not affect the prediction ability of the system because both the training and testing process are done by one specific gripper and the error between different sensors is also the same for all objects. This problem can be solved in the future by using more stringent and unified fabrication standards.

Additionally, for those grasps, some of them from different objects may look similar at a certain time frame (i.e., the time of taking photo, or the time of drawing the pressure maps). However, as we mentioned in main manuscript and above, we are using a period of output data (200 data points at time series) of 15 channels as a training sample, which means the data includes the information of the contact force, speed, sequences, contact positions, latency, and the contact durations etc. Those multi-dimensional features, rather than the individual grasping position, will then define the identity of the grasped object. As a result, as long as those objects have distinct shape differences,

the object recognition can be achieved.

Actual Change: We have added the detailed explanation in Text S5 in Supplementary Materials. Fig. R3 has also been added in the Supplementary Materials as Fig. S17.

Fig. R3. Peak voltage maps of the T-TENG sensors for 16 different objects.

Comment 3: I suggest the authors conduct an experiment where all data from 10-11 objects are used in training and all data from the rest 5-6 objects are used in testing. This will enable the research community understand the applicability of their sensors in more general object recognition, where generalisation to unknown objects is required.

Response:

We appreciate the reviewer's suggestion on the experiment of generalisation to unknown objects. Here we choose tennis ball, big box and short can to represent the three most common shapes of objects in daily life: sphere, cube and cylinder, as the unknown objects for testing. In the experiment, we still use the 100 grasp data of the rest 13 objects for training, therefore each raw data consists a period of output data in time series of 15 channels, 200 data points * 15 channels = 3000 features in total for each sample as the input to the machine. Again, we understand that the machine learning algorithm identifies objects still based on a complete set of 15 channel sensor

data in each grasp, i.e., hundreds of data points extracted from the 3000 data points of 15 channels by PCA for each sample data, not just the peak values.

For human being, it could be relatively easy to tell the difference when we look at the peak voltage map. Therefore, we conduct the next three experiments to verify effectiveness of general object recognition, in particular, the generalisation to unknown objects, under our sensor design and present machine learning algorithm.

(1) Tennis ball test

The first experiment is to see the recognition outcome from the machine for the unknown object like a tennis ball based on the previous training data which does not have tennis grasp data at all. We collected 100 grasp data samples for the tennis ball, i.e., the unknown object data for the trained machine model. Then we see the recognition outcomes for these 100 grasp data samples for the tennis ball, 51 samples are recognized as the baseball, and 40 samples are predicted as the apple. From the 51 samples, one of the sample data is selected and only the peak voltage value is identified and provided as the peak voltage map of the T-TENG sensors of tennis ball in the Fig. R4a. Then, among the 40 samples, one of the sample data is selected and the peak voltage map of the T-TENG sensors of tennis ball is identified as shown in the Fig. R4b. One of the typical training data of the baseball and the apple are selected and the peak voltage maps are shown in Fig. R4c and Fig. R4d. We try to leverage the peak voltage map to help us to figure out any relevant information for the machine to do the unknown object recognition.

The values and distribution of sensor outputs among channel 1-3, 7-9, 11 and 13 are quite similar among these three spherical objects. Furthermore, the output of channel 15 in Fig. R4a is close to the value in the Fig R4c, while the output of channel 15 in Fig. R4b is close to the value in the Fig R4d. This could be a factor observed by the machine and then suggest the sample data of R4a as the baseball, and the sample data of R4b as the apple.

Back to the point indicated by reviewer “This will enable the research community understand the applicability of their sensors in more general object recognition, where generalisation to unknown objects is required.”, I think the above results show that our sensors and machine algorithm works effectively to recognize the spherical object.

Fig. R4. Peak voltage maps of tennis ball samples predicted as a) the baseball and b) the apple respectively; peak voltage maps of c) the typical baseball sample and d) the typical apple sample.

(2) Big box test

The second unknown object recognition experiment is to test the big box grasp data for 100 times based on the trained model as same as the experiment 1. For the big box, most samples (total 70) are recognized as the orange, and only a few samples (21) are predicted as the baseball. From the 70 samples, one of the sample data is selected and the peak voltage map of the T-TENG sensors of big box is provided in the Fig. R5a. Then, among the 21 samples, one of the sample data is selected and the peak voltage map of the T-TENG sensors of big box is identified as shown in the Fig. R5b. One of the typical training data of the orange and the baseball are selected and the peak voltage maps are shown in Fig. R5c and Fig. R5d.

Although we have one cube shaped object, i.e, the small box, in the trained model, we do not get the recognition outcome of recognizing “big box” as the “small box”. The dimension of big box and small box are 10 cm * 4 cm * 14cm and 8 cm * 2 cm * 14cm, respectively, where the length of the big box is slightly longer than that of the small box, and the width of the big box is twice that of the small box. When we look at the 7 and 8 voltage maps in the Fig R3, the voltage value in channel 11 for the small box is obviously the highest among all channels, which is quite different from that of the big box where the voltage value in channel 11 is much smaller than that of the channel 9. The voltage distribution difference between the small box and the big box may finally make the machine recognize the big box as other existing objects in the training set with more similar voltage distribution.

On the other hand, when we look at the peak voltage maps in Fig R5, the values and distribution of sensor outputs among channel 1-3 and channel 11 of the big box shown in the Fig. R5a are quite similar to that of the orange shown in Fig. R5c. This could be a factor that most big boxes with such voltage distribution are identified as the orange. But for a few big box samples which have higher voltage values in channel 2-3 and 11 shown in the Fig. R5b, the baseball that also has

higher values in these channels (Fig. R5d) than that of the orange becomes the most possible prediction result of these samples.

After all, with the general understanding of machine learning, insufficient trained data in the cubic objects may make the machine suggest the 2nd best-matched objects (the orange) existing in the training set and putting more cubic objects of different sizes and types into the training set may be helpful to enhance the machine's ability to recognize the general cubic objects.

Fig. R5. Peak voltage maps of big box samples predicted as a) the orange and b) the baseball respectively; peak voltage maps of c) the typical orange sample and d) the typical baseball sample.

(3) Short can test

The third experiment is to identify the unknown short can grasp data for 100 times based on the trained model. Among these samples, more than half (52) are predicted as the long can, and 23 and 22 samples are recognized as the orange and the banana respectively. Among the 52, 23 and 22 samples, one of the sample data is selected from each category and the corresponding three peak voltage maps of the T-TENG sensors of the short can are provided in the Fig. R6a, Fig. R6b and Fig.R6c respectively. One of the typical training data of the long can, orange and banana are selected and the peak voltage maps are shown in Fig. R6d, Fig. R6e and Fig. R6f.

According to the peak voltage maps in Fig. R6a and Fig. R6d, it is clear that the short can and the long can have almost the same peak voltage distribution among the 12 T-TENG sensors, where peak voltage values in channel 1-3, channel 9, channel 11, channel 13 and channel 15 are usually larger than the remaining channels and the range is also relatively stable, which may make the recognition result of the long can dominates. For the few samples of short can that have relatively low values in channel 4-6 shown in the Fig. R6c, the typical banana sample which also has small peak voltages in these channels in the Fig. R6f shows a higher similarity to these samples and this may be the decisive factor that the sample data of Fig. R6c is recognized as the banana. However, for the sample data visualized in Fig. R6b and Fig. R6e, it's difficult to directly tell the similarity

just based on the peak voltage maps and the machine may make the prediction based on more features with higher dimensionality in the raw data, not just the peak voltage values.

The above results show that our sensors and machine algorithm also work effectively to recognize the cylindrical object even though there are only a small number of cylindrical samples in the training set.

Fig. R6. Peak voltage maps of short can samples predicted as a) the long can, b) the orange and c) the banana respectively; peak voltage maps of d) the typical long can sample, e) the typical orange sample and f) the typical banana sample.

In summary, according to the prediction results of unknown objects given by the pre-trained SVM model based on the raw data (3000 data points) of samples, as well as the referential analysis of the voltage maps, the above results for three unknown objects indicate that our developed sensors have a certain recognition ability when encountering general spherical and cylindrical objects, which will be highly likely to be identified as objects with similar shapes in the training data set. While in the experimental 2, due to insufficient trained data in the cubic objects, we realized that machine may suggest the 2nd best-matched results because of the relative values and distribution observed in the outputs of the 12 T-TENG sensors. The prediction ability of our system for general cubic objects can be enhanced in the future if we put more cubic objects of different sizes and types into the training set and make the machine learn more unique features of cubic objects.

It is worth to mention that, SVM technique as a supervised learning algorithm, all of the recognizable objects were labelled during the training process. Hence, if we introduce the unknown object for testing without training, then SVM will not be able to provide the true identity of itself (since it does not exist), but to categorized it into the trained labels which have the most similar features. In another word, for unknown objects, the function of SVM-based finger grasping becomes classification (based on the existed classes, i.e., the trained 10~11 objects), rather than the precise recognition (i.e., provide the true identity of itself), and the classification should be shape, position, and size-based recognition. Hence, that is the reason for above test results of the unknown objects recognition, i.e., the tennis ball is categorized into the baseball or the apple, and the short can is identified as the long can. Moreover, it is universally accepted that the performance of generalization also relies on the population (classes) of trained data set, and hence,

for larger database, there will be more chance to find the similar object which matches the unknown object better. For the above example of big box test, if we can establish a data base with more trained objects which contains other boxes with different sizes or shapes, the recognition performance will then be effectively improved.

Reference:

1. Abdi, H. & Williams, L. J. Principal component analysis. *WIREs Comp Stat* 2, 433–459 (2010).
2. Burges, C. J. C. A Tutorial on Support Vector Machines for Pattern Recognition. *Data Mining and Knowledge Discovery* 2, 121–167 (1998).

REVIEWERS' COMMENTS

Reviewer #1 (Remarks to the Author):

I would like to thank again the authors for their detailed experiments, and for analytically addressing my comments.

The PCA and SVM section is now more complete and helps the reader understand how the data were gathered, processed, and utilised.

The new experiments are important because they demonstrate both the strengths and the limitations of the TENG sensor in object recognition. The TENG signals are similar for rounded surfaces, leading to good recognition of curved surfaces (sphere and cylinder recognition experiment), and additional research can improve the results in flat surface recognition.

I suggest publication of the article, and I wish the authors further success in their future research.

Reviewers' comments (R1):

Reviewer #1 (Remarks to the Author):

Paper Description

This paper describes two novel low-cost sensors for robotic perception, based on triboelectric nanogenerators (TENGs). In the introduction, the authors describe the need for smarter, low cost and high-information sensors for factories and industrial digital twins. They mention some disadvantages of modern sensors for robots with rigid fingers, and some of the sensors used in soft robotics. They then describe some methods of sensing in soft robotics (piezoelectricity, triboelectricity e.a.) and the existing use of TENGs in robotic sensing, combined with glove-based user-input devices. Their main thesis is that their developed sensors fill the gap in the sense that they are cheap to produce, convey large amounts of information, and contrary to exiting works they can detect contact, slippage, and contact type (point or area contact). They then describe their sensors: the length-TENG (L-TENG) sensor consists of a positive triboelectric component, which is a gear with four teeth. The negative component is an elastic strip of PTFE. As the strip is pulled, the gear's teeth momentarily touch the strip and result in the generation of a brief triboelectric voltage spike. The distance between spikes and their number, carry information related to the gear spinning angle and direction. A spring then ensures the smooth transition of the strip to the initial state. The tactile-TENG (T-TENG) sensor consists of five electrodes, one along the edge of a strip, and four across different points. Through triboelectric and electrostatic induction effects, the electrodes fire with different amplitudes when there is a contact. By comparing the electrode outputs, as well as the time difference between them, the sensor can detect contact and sliding motion along its length. The amplitude of the electrode along the edge provides info about the type of contact. The authors then demonstrate the usefulness of the sensors in a series of experiments. They control the bending angle and direction of a robotic finger with a glove that houses the two sensors. They then present a custom pneumatic soft robotic finger that can house the T-TENG sensor, calibrate it with and test the finger-sensor integration and the sensor output profiles by grasping a number of objects (apple, orange egg and turnip). They also show how the sensor signals extracted from grasping various objects (cube, cylinders, spheres and ring) can be used in tactile object recognition through SVM learning, achieving high recognition accuracy. Finally, they employ the SVM algorithm in a digital twin factory scenario, where a real soft gripper equipped with the T-TENG sensor recognizes various objects, and the digital twin robot sorts the object in different objects.

Paper strengths

Overall, the paper is well written. The introduction is well formed, and the authors incrementally build their arguments towards the novelty of their sensors and potential uses, and explain the merits compared to the existing literature. I believe that both the L-TENG and T-TENG sensors show very innovative designs, and the authors do a remarkable job in explaining how the output voltage signals are generated, and how these voltage profiles can be used to express robotic joint outputs and to detect slippage and contact detection. The experiments presented are simple and "to the point", clearly explaining their purpose and proving the arguments. The HMI control experiment provides a fundamental mapping of the

L-TENG sensor output to a robotic joint position. I feel that the experiment would be more representative of robotic HMI control if the authors could show the continuous movement of the L-TENG on the HMI and the corresponding motion of the robotic joint, instead of 30-degree intervals, and I strongly encourage them to do so. The soft finger experiment helped showcase how the voltage profiles of the T-TENG sensor are used in object grasping. The object recognition experiment achieves very high accuracy, and I would like to see the results of it extended to new objects of varying curvatures (the YCB dataset would be a good test case [1]).

Paper shortcomings

The main criticism I have to offer is that the paper feels cramped. The authors try to present and solve a large range of problems related to joint position sensor design, tactile sensor design, robot teleoperation with HMIs, soft finger design, soft finger grasping, object recognition and learning, and digital twin applications. This is reflected to both the introduction section, where the authors list too many potential applications to solve, as well as the experimental methodology (i.e. lots of small-scale experiments to test a broad variety of issues). I believe that the digital twin experiment, albeit successful, was a bit unnecessary as it merely showed a grasping simulation executed with the SVM results as input. I would suggest that the paper could be broken into two papers: one sensor-oriented, that would describe analytically the sensors' principle of operation, the voltage profile generation, and the HMI control with more elaborate control methods, and one application oriented, where the authors could pick up a potential application (learning, digital twin or other) and demonstrate the usefulness of the sensors through thorough experiments and comparison with existing solutions of their chosen problems (for example, how does their T-TENG sensor perform compared to tactile sensors and skins in an object recognition experiment?).

Minor issues

The authors should check the spelling of some words throughout the manuscript and title.

Comments on the supplementary material

The supplementary material and schematics were useful to understand the soft finger operation. The video of the digital twin execution is simple and indicative. The video of the HMI is also clear, however it left me confused over the use of the T-TENG sensor. The authors claim in the paper that the T-TENG sensor is used to change the bending direction, however in the video it looks as if the T-TENG sensor does not affect the motion direction (the bending direction just seems to copy the motion of the glove finger). The authors should clarify a bit more how the T-TENG sensor is used on the HMI control experiment, and show it with some schematics or images.

References

[1] Calli, B., Singh, A., Bruce, J., Walsman, A., Konolige, K., Srinivasa, S., Abbeel, P. and Dollar, A.M. Yale-CMU-Berkeley dataset for robotic manipulation research. *The International Journal of Robotics Research*, 36(3), pp. 261-268 (2017).

Reviewer #2 (Remarks to the Author):

The manuscript enabled “Development of Low-cost Human-Machine-Interface Using Triboelectric Nanogenerator Sensors for Future Digital Twin” reported a self-powered intelligent warehouse system integrated with triboelectric nanogenerator (TENG) technology. Due to the synergistic effect of a length TENG (L-TENG) and a tactile-TENG (T-TENG), this smart system trained by machine algorithm exhibited an impressive capability of object identification with decent accuracy. Generally, this work is well illustrated with clearly organized figures and strong logics. The only concern is that this manuscript is mainly a fancy engineering technology demonstration, where TENG technology was used as a sensing interface to achieve artificial intelligence/machine learning. This application itself is not quite new, and there is almost no new fundamental understanding, new mechanism illustration, materials innovation, or novel discovery. The scientific value of this work is not significant. Again, it is a nice piece of engineering demonstration. It will be very helpful if the authors could elaborate more about technology advancements achieved in this work. Other minor concerns listed below should also be addressed:

1. Page 3, line 63-65, in the introduction section, the authors claimed that resistive sensors suffer from temperature instability that prevents their practicality. However, temperature also impacts TENG output according to others’ report (Advanced Engineering Materials 19.12 (2017): 1700275.). In order to show the superiority of TENG over other technologies, the authors are suggested show the accuracy of object identification under different temperature.
2. Page 4, line 88-90, the statement that soft materials of TENG and PENG has the same Young’s modulus level as silicone and TPU is not correct. Piezoelectric soft polymer such as PVDF has Young’s modulus up to 1 GPa, which is a few orders of magnitude higher than the modulus of soft silicone and TPU (1-10 MPa).
3. The measurements shown in Fig 3 are qualitative, such as tapping in the middle area between two electrodes will trigger smaller signals compared to direct tapping above the electrodes, and more contact area, larger output. This is all empirical. The contact speed and strain speed will all affect the output. Is there any fitting equation/curve could bridge position and output to predict the precise output at different position under different contact and strain speed? This might help further improve the accuracy.
4. How is the stability of the smart system? After cycles of utilization, is the accuracy still over 97%?
5. The authors should check typos before submission. The typos such as “nanaocomposites” in page 4, line 86 should be corrected

Reviewer #3 (Remarks to the Author):

The major claims of the paper include the design of a flexible sensor called a L-TENG sensor to capture continuous motion of a soft gripper coupled with a T-TENG sensor for tactile sensing. The reviewer has listed several comments that will enhance the quality of the submission.

1. The work presented will be of interest to the audience of the journal and appears to be an extension of previous TENG advancements. The work described in the text along with the supplemental information indicate that the work can be reproduced by the authors.

2. The authors state, ". Compared to the previous solutions, the developed sensing system with patterned-electrode tactile TENG (T-TENG) sensor can detect sliding, contact position and gripping mode. " This statement would be better supported with quantitative comparison of the proposed new design with other solutions, optical etc. in a small table or figure.

3. The authors state, "With a layer of silicone rubber, a kind of flexible and stretchable negative triboelectric material, coated on the surface of the patch, triboelectric signals can be generated by the stimuli on the T-TENG sensor's surface and collected in these five electrodes," and couple this with a methods section. The process could be described in a more detailed manner and paired with the appropriate figure.

4. Since sensor and fabric application to the host influences detection and output signal, brief explanation of the method/type of adhesive used would be beneficial.

5. Figure 1 is difficult to read. In particular, the fonts are too small to read. Also, the relationship between elements in the figure are not clearly illustrated or described in the caption/text.

In summary, the work described is interesting and a meaningful advancement in TENG technology that can be applied to many applications.

Manuscript number: NCOMMS-20-04528A

Title: Triboelectric Nanogenerator Sensors for Soft Robotics Aiming at Digital Twin Applications

Authors: Tao Jin, Zhongda Sun, Long Li, Quan Zhang, Minglu Zhu, Zixuan Zhang, Guangjie Yuan,

Tao Chen, Yingzhong Tian, Xuyan Hou and Chengkuo Lee

Responses to the reviewers:

Reviewer #1

Comments 1: I feel that the experiment would be more representative of robotic HMI control if the authors could show the continuous movement of the L-TENG on the HMI and the corresponding motion of the robotic joint, instead of 30-degree intervals, and I strongly encourage them to do so.

Response:

We thank a lot for reviewer's valuable comment and suggestion. In terms of continuous motion sensing, there are several researches have demonstrated the continuous finger bending sensor, such as the grating patterned TENG sensor (Nano Energy, 2018, 54, 453–460), and many resistive based sensors (ACS Sens. 2016, 1, 817–825), etc. For our proposed L-TENG, the fundamental working principle is able to realize the continuous sensing. It has a resolution of 5mm. For the data glove presented in our paper, the length variation is limited by the finger deformation when the PTFE strip of L-TENG is pulled out by finger, and the number of contacts between the gear teeth and the strip is then limited for representing the displacement. Therefore, we realized the 30-degree interval. It is possible for higher accuracy as we mentioned in Page 11 (i.e., increasing transmission ratio or the number of the gear tooth, etc.). The L-TENG with corresponding structure for improving precision is planned in our future work. To make it feasible, we can apply fabrication process with high precision, such as metallurgy or laser machining, in order to create fine pattern of teeth to enhance the resolution.

On the other hand, as our main novelty in this manuscript, we proposed tactile sensor (T-TENG) and length sensor (L-TENG) to form a perception system with machine learning technique for improving the intelligence of the soft gripper. **The peak counting and readout of output ratio are the data processing strategies which can eliminate the absolute amplitude variations of triboelectric output caused by environmental variations, and hence, to ensure the signal stability throughout the usage.** Moreover, by effectively utilizing the multi-dimensional data provided by two different sensors, the machine learning approach can provide equivalent precise measurement compared to continuous sensing regarding the object recognition, while avoiding the huge amount of data acquisition for continuous monitoring.

Comments 2: I would like to see the results of it extended to new objects of varying curvatures (the YCB dataset would be a good test case [1]).

Response: We appreciate the reviewer's suggestion on extending objects of varying curvatures. We

agree that it would be better if more objects can be applied. In our previous work, only one T-TENG and one L-TENG sensor were applied to collect the data of 6 objects and we tested 200 times for machine learning training. It is feasible for our system to further enhance the recognition capability by using more sensors (each finger has one T-TENG sensor and one L-TENG sensor). Therefore, in response to the reviewer's advice, 16 types of objects with various shapes have been tested to construct the new data set, where we used 15 sensing channels (four channels of T-TENG and one channel of L-TENG on each of 3 soft actuators). Our result as depicted in Fig. 5 indicates that when the number of the objects was increased to 16, the recognition accuracy can still reach up to 98.1% for all the 16 types of objects with 15-channel signals. Besides, the model trained by data contracted from the previous data set showed that more applied samples can provide the better result (Page 19). Hence, we believe that our system can be used to recognize more objects of varying curvatures with the aid of introducing more sensor channels.

In the meantime, it is also worth to mention, in the practical application, such as fruit sorting in a specific company, the number of training objects for each job task is usually remain at few tens level. Hence, it will not be necessary for maintaining such high adaptability to all of the unnecessary objects in a certain scenario (i.e., unnecessary "pen" recognition for fruit sorting). In another word, for a small task including 5~10 objects, the minimalistic design with only one finger equipped with sensor is enough. For more complex tasks, we can then add extra sensors on other fingers to enable the scalability.

Actual change: The detailed content can be found in the Data Processing via Machine Learning Technology section from Page 16-20. Fig. 5, Fig. S14, Fig. S15, and Movie S5 were added to illustrate the concerned work.

Comments 3: The main criticism I have to offer is that the paper feels crammed. The authors try to present and solve a large range of problems related to joint position sensor design, tactile sensor design, robot teleoperation with HMIs, soft finger design, soft finger grasping, object recognition and learning, and digital twin applications. This is reflected to both the introduction section, where the authors list too many potential applications to solve, as well as the experimental methodology (i.e. lots of small-scale experiments to test a broad variety of issues).

Response: We thank a lot for reviewer's valuable comment and suggestion. To clarify our main idea in this manuscript, we propose a triboelectric nanogenerator (TENG) based smart sensory system for soft robotics, which consists of length TENG (L-TENG) and tactile TENG (T-TENG) sensors. Both of two types of sensors based on TENG technology were designed to solve the sensory problems of the soft gripper for digital twin applications. In Fig. 2 and 3, the presented results offer the basic characterizations of two sensors for understanding the working mechanisms and verifying the functionality with the real-time signal processing via a simple demonstration of glove-based sensing. Afterwards, we focused on the integration of the sensors into the soft gripper for enhancing the intelligence with the aid of machine learning algorithm, in order to realize the object recognition for digital twin application. In Fig. 5e-g, in response to reviewer's question about the necessity of combining several concepts, the results shown the importance of the fusion of two types of sensors by improving the recognition accuracy significantly, as each of two sensors offers the corresponding sensing dimension to increase the available features to be extracted for machine learning. In general, the digital twin is the potential application, and the soft finger is the main platform to perform digital twin. But it needs sensory system to achieve intelligent digital twin, which requires the joint position

sensor (L-TENG) design and tactile sensor (T-TENG) design, while the robot teleoperation is just served as a preliminary verification of sensors' functionalities. In addition, we have rewritten our introduction and rearranged the sections of the manuscript for easier understanding where some content about the HMI has been delete.

Actual changes: The second and third paragraph of the introduction have been modified for expressing the necessity for developing soft robotic sensing and its background better. All the change has been highlighted from Page 3 to Page 5 in the manuscript and hope to dissipate reviewer's concern.

Comments 4: I believe that the digital twin experiment, albeit successful, was a bit unnecessary as it merely showed a grasping simulation executed with the SVM results as input. I would suggest that the paper could be broken into two papers: one sensor-oriented, that would describe analytically the sensors' principle of operation, the voltage profile generation, and the HMI control with more elaborate control methods, and one application oriented, where the authors could pick up a potential application (learning, digital twin or other) and demonstrate the usefulness of the sensors through thorough experiments and comparison with existing solutions of their chosen problems (for example, how does their T-TENG sensor perform compared to tactile sensors and skins in an object recognition experiment?).

Response: Thank you for reviewer's kind advice to our work. We apologize for the confusion of the main concept intended to present. As mentioned above, our main purpose is to develop a sensory system specifically for digital twin and capable of conducting object recognition with enough accuracy, which highly relies on the fusion of two types of sensors for soft robots to acquire enough sensible dimensions. All the sensor designs and technologies mentioned in the text are used for implementing an intelligent smart gripper system. In contrast, the presence of single type sensor will bring a severe decline of recognition accuracy and make the whole system unreliable (Fig. 5f and 5g). The glove-based demonstration is only served as a simple verification to show the feasibility of real-time signal processing for further researches on digital twin. Then, we tested the soft gripper integrated with designed sensors for robotic sensing. Finally, we showed the application of the soft gripper with the sensors for object recognition and concerned digital twin. This system has great potential for unmanned warehouse, farm or shop floor, while there is no light for camera sensing. Based on reviewer's comments, we have obtained a new data set by testing 16 types of objects with 15 channels (four for T-TENG and one for L-TENG on each soft actuator) and then trained the machine learning model by activating the data of L-TENG channels (3 channels), T-TENG channels (12 channels), and all the 15 channels respectively. The results indicate that the model trained by 15-channel data shows the best accuracy (98.1%) than the others. Meanwhile, the model trained by T-TENG channels' data performs better than the L-TENG's, especially for the object with similar curvature. Besides, we contracted the data of the previous data set and found that the model trained by 100 samples performs lower accuracy than the previous model. Therefore, we can conclude that the more sensory information, the more accurate the model will be and the sensory data from the T-TENG seems more important for object recognition. However, both the T-TENG sensor and L-TENG sensor contribute to the overall accuracy of the system.

Based on reviewer's comment, we also made some comparisons about the existing sensors for soft robotics by TENG technology. Prof. Wang's group firstly presented a soft robot with the TENG tactile sensor for proximity and pressure detection (Adv. Mater., 2018, 30, 1801114). They discussed the contact position for robotic clawer by comparing the signals in different electrodes. In reference (Adv.

Mater. Technol., 2019, 1900337, 1–7), the soft gripper was covered by the TENG tactile sensor with only one electrode. They presented the signals as the gripper gripped or released the objects. In addition, Chen, S. *et al.* presented a wire-driven soft gripper with three TENG tactile sensors (Adv. Mater. Technol., 2020, 1901075, 1–10) to detect the contact and compared the signals after gripping different objects. These existing solutions mainly concentrate in the view of the contact process or status (i.e., proximity, contact, gripping, and release) by evaluating the signals of single or multiple sensors. These sensors work separately and thus it is hard to perceive the accurate position (only right above the electrodes) or contact surface.

Actual Change: The detailed content can be found in the Data Processing via Machine Learning Technology section from Page 16-20. Fig. 5, Fig. S14, Fig. S15, and Movie S5 were added to illustrate the concerned work.

Comments 5: The authors should check the spelling of some words throughout the manuscript and title.

Response: We apologize for these mistakes in our manuscript. We have checked the spelling again and marked the modification.

Actual changes: The concerned content has already been highlighted throughout the manuscript.

Comments 6: The supplementary material and schematics were useful to understand the soft finger operation. The video of the digital twin execution is simple and indicative. The video of the HMI is also clear, however it left me confused over the use of the T-TENG sensor. The authors claim in the paper that the T-TENG sensor is used to change the bending direction, however in the video it looks as if the T-TENG sensor does not affect the motion direction (the bending direction just seems to copy the motion of the glove finger). The authors should clarify a bit more how the T-TENG sensor is used on the HMI control experiment, and show it with some schematics or images.

Response: We are sorry that the T-TENG in our manuscript is not well-explained. For verifying the real-time signal processing system, we use a slice of T-TENG with single electrode to switch the motion direction of the robotic hand and it is mounted on the thumb position of the glove, which has been mentioned in Page 12. We have modified the Fig. 3f(i) to express the location of the T-TENG better. As depicted in Fig. 3f(vi), the motion direction of robotic finger will be changed after the trigger caused by the tapping of the thumb and index finger. Besides, the position of the both sensors has been marked in the supplementary movie (Movie S3). We hope these modifications may dissipate review's confusion.

Actual changes: The concerned content can be found in Page 12, which has been highlighted. And the modified movie can be found in supplementary material (Movie S3).

Reviewer #2

Comments 1: This application itself is not quite new, and there is almost no new fundamental understanding, new mechanism illustration, materials innovation, or novel discovery. The scientific value of this work is not significant.

Response:

We thank a lot for reviewer's valuable comment. In this paper, we proposed a tactile sensor (T-TENG), a length sensor (L-TENG), and a perception system with machine learning technique for improving the intelligence of the soft gripper. To indicate the scientific value, there are several points which can be

clarified.

1. We integrated the triboelectric sensor on pneumatic finger to realize self-powered sensing, which can reduce the power consumption for massive deployment, especially in wireless AIoT framework. There are only a few papers studied triboelectric based sensing for pneumatic actuator or continuous motion sensing. But they do not possess the capability of multi-dimensional sensing for practical application, especially digital twin.
2. We firstly proposed the gear-based length sensor (L-TENG) to measure the bending degree of the soft gripper, which has the specially designed disc spring for recovery after air off for the soft actuator. The special gear-based structure of the L-TENG sensor has not been proposed before and can be used for continuously sensing of the displacement which is difficult to be achieved by common triboelectric based sensors. The resolution can also be further improved as we mentioned Page 11 by adjusting the transmission ratio or gear tooth number, etc.
3. T-TENG can not only detect the siding, contact position but also contact area with simple structure, which is useful to distinguish surface contact and point contact for soft gripper.
4. Refer to Fig. 2, we proposed the special design of T-TENG electrodes with the readout method of output ratio. Together with peak counting readout of L-TENG shown in Fig 3, we can solve the problem of output amplitude fluctuation caused by environmental noises.
5. We successfully utilized the constant curvature theory to explain the bending angle of pneumatic finger by the outputs from TENG sensor, and obtained the results shown in Fig. 4c. This approach verifies the feasibility of realizing primary recognition without advanced data processing.
6. For more complex tasks in practical application, we investigated the object recognition assisted by the machine learning technique. To achieve this task, we effectively combined L-TENG and T-TENG to create the enough dimensions of sensing data for increasing the recognition accuracy, as shown in Fig 5.
7. The proposed sensory system should have potential for unmanned warehouse, farm land and shop floor management, where there is no light or no camera, as shown in our integrated demonstration.
8. The object recognition together with sensory system also helps to build up a comprehensive database by the continuous accumulation of sensing data during million times of grasping of diverse objects in practical application. These collected massive data become a valuable guideline and evaluation about the performance of grabbing object by robots, such as the contact mapping which reflects the most frequent touched areas or less touched areas, as well as the slippery area. This information is crucial for those robotic arm engineers to improve the designs accordingly.

Comments 2: Page 3, line 63-65, in the introduction section, the authors claimed that resistive sensors suffer from temperature instability that prevents their practicality. However, temperature also impacts TENG output according to others' report. In order to show the superiority of TENG over other technologies, the authors are suggested show the accuracy of object identification under different temperature.

Response: Thank you for reviewer's kind advice. We agree with the reviewer's opinion that the working temperature may affect the object identification. However, it usually happens at very high or low temperature. As mentioned in some literature data (Nano Energy, 2014, 4, 453–460; Appl. Phys. Lett., 2015, 106, 013224), the triboelectric output will experience the significant fluctuation after reach above 320K or below 200K. Hence, the triboelectric based sensor can provide stable functions under the practical scenarios, where the temperature seems to remain in a constant value (298K) and the

fluctuation of temperature is quite small. As the fundamental principle of machine learning algorithm is to identify the signal patterns and extract the distinct features among various objects. Hence, for a specific object, the system can recognize it successfully for each cycle if the output of contacting can maintain the consistency of the signal patterns, which means the high accuracy. As reviewer's kind advice, we have tested the signals for some of the objects in diverse temperatures (298K and 313K). As depicted in Table 1, our results indicate that within normal ambient temperature, our TENG sensory system is not very sensitive to the temperature. In addition, we can also improve the accuracy of the model by getting more data under different temperatures for training to enhance the versatility of the model when the environment has large temperature variation.

Actual Change: Table 1 in reply can also be found in the Supplementary Materials (Table S2)

Table 1 The real-time object recognition results under different temperatures

Temperature	Small box	Orange	Apple	Long Can	Short Can	Accuracy
298K	20/20	19/20	19/20	20/20	19/20	97/100 (97%)
313K	20/20	17/20	18/20	19/20	20/20	95/100 (95%)

Note: "19/20" means that 19 times correct recognition in 20 times gripping tests.

Comments 3: Page 4, line 88-90, the statement that soft materials of TENG and PENG has the same Young's modulus level as silicone and TPU is not correct. Piezoelectric soft polymer such as PVDF has Young's modulus up to 1 GPa, which is a few orders of magnitude higher than the modulus of soft silicone and TPU (1-10 MPa).

Response: We apologize for the mistake in our manuscript. Sensors made of PVDF film can also show good flexibility if it is thin enough, although it seems hard to be stretched. Therefore, thin PVDF film can also be used in soft gripper if it has a stretching limitation layer to make the soft actuator bend. Thanks for reviewer's advice and we have modified the corresponding expression.

Actual Change: The concerned content can be seen in Page 3.

Comments 4: The measurements shown in Fig 3 are qualitative, such as tapping in the middle area between two electrodes will trigger smaller signals compared to direct tapping above the electrodes, and more contact area, larger output. This is all empirical. The contact speed and strain speed will all affect the output. Is there any fitting equation/curve could bridge position and output to predict the precise output at different position under different contact and strain speed? This might help further improve the accuracy.

Response: We would like to express our most gratitude to the reviewer's kind advice. For distinguishing the signal caused by different contact and strain speed, we used the formula

$$Ratio = E_i / \sum_{i=1}^4 E_i$$

as mentioned in Page 9, which presents the ratio of the output value of the single electrode to all of the four short electrodes.

Fig. R1 The simplified diagram of the contact process for electrostatic analysis.

The main working mechanism is listed as following. According to the electrostatic induction, we can define the electric potential as

$$U = k \frac{Q}{r} \quad (1)$$

where Q is the amount of charge, r is the distance to the point charge, and k is Coulomb's constant as shown in Fig. R1 below. So, if we assume that a charge of $+Q$ moves to the silicone rubber surface with a distance of h , the output voltage of two nearby electrodes can be expressed as

$$\begin{cases} V_{E_i} = k \frac{Q}{\sqrt{x^2 + h^2}} - k \frac{Q}{x} \\ V_{E_{i+1}} = k \frac{Q}{\sqrt{(l-x)^2 + h^2}} - k \frac{Q}{l-x} \end{cases} \quad (2)$$

where x represents the distance between the touch point to E_i and $l-x$ represents the distance between the touch point and E_{i+1} . Thus, the ratio can be derived as

$$\frac{V_{E_{i+1}}}{V_{E_i}} = \frac{\frac{1}{\sqrt{(l-x)^2 + h^2}} - \frac{1}{l-x}}{\frac{1}{\sqrt{x^2 + h^2}} - \frac{1}{x}} \quad (3)$$

So if h is large enough, we can obtain

$$\frac{V_{E_{i+1}}}{V_{E_i}} = \frac{x}{l-x} \quad (4)$$

Therefore, the ratio can be used to explain the method to get a more accurate position under virus speeds and strain speeds. Similarly, as the electrodes number are increased to four, we can use

Ratio = $E_i / \sum_{i=1}^4 E_i$ to sense the position. As depicted in Fig.2d, we analyzed the data after contacting

right above the electrode and the middle position of two electrodes. The result shows the reliability of the T-TENG to sense the contact position.

Actual Change: The concerned content can be seen in Page 9 and we have added the detailed explanation in Text S1 in Supplementary Materials.

Comments 4: How is the stability of the smart system? After cycles of utilization, is the accuracy still over 97%?

Response: We would like to express our most gratitude to reviewer's effort and patience in reviewing

our manuscript. According to the comment, we have used the soft gripper integrated with both L-TENG and T-TENG sensors (16 channels) to grip various objects with 2000 times to show the stability of our sensory system. In order to reduce the proportion of necessary training and testing times in the entire stability test and make the result for certain utilization cycles more accurate, we choose to use less data: 80 samples for training for each object (total 5 objects) and 20 samples for testing for each object after cycles of utilization. As shown in Table 2 below, our results indicate that though the recognition accuracy has some inevitable fluctuation due to the environmental and man-made noise, the average accuracy remains high level and there isn't an obvious decrease after gripping for 2000 times. Therefore, we believe that our design can keep a high accuracy even though after cycles of utilization.

Actual changes: Table 2 in reply can also be found in Supplementary Materials (Table S3)

Table 2 The test results for verifying the stability of the smart system

	Gripping Counts								
	100	300	500	700	1000	1300	1500	1800	2000
Small Box	20/20	20/20	20/20	19/20	20/20	18/20	20/20	20/20	20/20
Orange	19/20	18/20	19/20	18/20	17/20	18/20	17/20	19/20	19/20
Apple	19/20	19/20	19/20	20/20	20/20	19/20	19/20	19/20	19/20
Long Can	20/20	20/20	18/20	19/20	18/20	20/20	20/20	20/20	19/20
Short Can	18/20	20/20	20/20	20/20	19/20	20/20	19/20	19/20	19/20
Accuracy	96/100 (96%)	97/100 (97%)	96/100 (96%)	96/100 (96%)	94/100 (94%)	95/100 (95%)	95/100 (95%)	97/100 (97%)	96/100 (96%)

Note: "19/20" means that 19 times correct recognition in 20 times gripping tests.

Comment 5: The authors should check typos before submission. The typos such as "nanaocomposites" in page 4, line 86 should be corrected.

Response: We apologize for these mistakes in our manuscript. We have checked the spelling again and marked the modification.

Actual changes: The concerned content has already been highlighted throughout the manuscript.

Reviewer #3

Comment 1: "Compared to the previous solutions, the developed sensing system with patterned-electrode tactile TENG (T-TENG) sensor can detect sliding, contact position and gripping mode." This statement would be better supported with quantitative comparison of the proposed new design with other solutions, optical etc. in a small table or figure.

Response: Thanks for reviewer's kind advice to our work and sorry for our mistakes. In this paper, we want to introduce the sensing methods for soft gripper which requires special design for sensors. Actually, we want to express that compared to the previous solutions based on TENG technology for robots, our device can detect sliding, contact position and gripping mode. The concerned sentence has been modified (Page 4). Meanwhile, the T-TENG sensor featuring simple structure and low cost has its advantages compared to other methods. As reviewer's kind comment, some comparisons about the tactile sensor based on TENG technology and other methods (e.g., piezoelectricity) have been listed in Tab.1 below.

Actual changes: The comparison of the tactile sensors can be found in Table 3 below (also see Table

S1 in Supplementary Materials). The concerned content has been modified in Page 4.

Table 3 Comparison of the tactile sensor based on TENG technology and other methods

Method	Position	Sliding	Contact Surface	Force	Electrode Number	Application
Triboelectric	Simple (Contact or separation)	No	No	No	One	Soft gripper ¹⁻³
Triboelectric	5 × 5 sensory array	Yes	No	No	Four	Robot control ⁴
Triboelectric	8 × 8 sensory array	Yes	Yes	No	Sixteen	Wearable device ⁵
Triboelectric	Simple (Contact or separation)	No	No	No	Two or three	Wearable device ^{6,7}
Capacitive, Triboelectric	4 × 4 sensory array	Yes	No	Normal	Eight	Humanoid Hand ⁸
Piezoresistive	No	No	Yes	No	One	Soft gripper ⁹
Piezoresistive	Yes	No	No	Yes	One	Soft gripper ¹⁰
Piezoresistive	Yes	No	Yes	No	Two or more	Soft gripper ¹¹
Piezoelectric	No	Yes	No	No	Two	Textile detection ¹²
Piezoelectric	No	No	No	Contact or not	One	Wearable device ¹³
Capacitive	Yes	No	No	No	Four	HMI ¹⁴
Capacitive	No	No	No	3-axis	Four	Humanoid Hand ¹⁵
Capacitive	Yes	No	No	Contact or not	Six	Wearable device ¹⁶
Thermosensitive	No	No	No	Normal	One	Humanoid Hand ¹⁷
Optical	Yes	No	No	Normal		Sensor Mechanism ¹⁸

Comments 2: The authors state, "With a layer of silicone rubber, a kind of flexible and stretchable negative triboelectric material, coated on the surface of the patch, triboelectric signals can be generated by the stimuli on the T-TENG sensor's surface and collected in these five electrodes," and couple this with a methods section. The process could be described in a more detailed manner and paired with the appropriate figure.

Response: Thanks for reviewer's kind advice to our work. We have added a figure about the fabrication process in Supplementary Materials (Fig. S1), which may do some helps. Besides, the methods part has been added some detailed parameters about the fabrication of the T-TENG sensor. Hope to dissipate the reviewer's concern

Actual changes: These changes can be found in the Methods section (Page 22) and the latest supplementary materials (Fig. S1).

Comment 3: Since sensor and fabric application to the host influences detection and output signal, brief explanation of the method/type of adhesive used would be beneficial. **Response:** We appreciate

the reviewer's suggestion on clarifying the method/type of adhesive used in our manuscript. At first, we used the PET film as the substrate to test the T-TENG's working mechanism. The conductive textile tape was pasted on the PET film by its own adhesive, a kind of acrylic pressure sensitive adhesive. Then, the EcoFlex 00-30 was used to cover and seal the Ni-fabric layer by a mold (see Fig.S1 in Supplementary Materials). Similarly, the same method has been used in the integration of T-TENG and the soft actuator as illustrated in Fig. S13 where the extra silicone rubber was used to reinforce the adhesion of the EcoFlex and TPU.

Actual Changes: The methods for fabrication T-TENG have been modified based on reviewer's advice Page 22. Besides, Fig. S1 has been added and the Fig. S13 has been modified for better explanation.

Comments 4: Figure 1 is difficult to read. In particular, the fonts are too small to read. Also, the relationship between elements in the figure are not clearly illustrated or described in the caption/text.

Response: We apologize for these mistakes in our manuscript. The concerned figure has been modified according to reviewer's advice. Thank you for reviewer's kind advice

Actual changes: These changes can be found in Fig. 1 of the latest manuscript.

Reference:

1. Chen, J., Chen, B., Han, K., Tang, W. & Wang, Z. L. A Triboelectric Nanogenerator as a Self-Powered Sensor for a Soft-Rigid Hybrid Actuator. *Adv. Mater. Technol.* 1900337, 1–7 (2019).
2. Lai, Y. et al. Actively Perceiving and Responsive Soft Robots Enabled by Self-Powered, Highly Extensible, and Highly Sensitive Triboelectric Proximity- and Pressure-Sensing Skins. *Adv. Mater.* 1801114, 1–12 (2018).
3. Chen, S., Pang, Y., Yuan, H., Tan, X. & Cao, C. Smart Soft Actuators and Grippers Enabled by Self-Powered Tribo-Skins. *Adv. Mater. Technol.* 1901075, 1–10 (2020).
4. Chen, T. et al. Triboelectric Self-Powered Wearable Flexible Patch as 3D Motion Control Interface for Robotic Manipulator. *ACS Nano* 12, 11561–11571 (2018).
5. Dong, K. et al. A Stretchable Yarn Embedded Triboelectric Nanogenerator as Electronic Skin for Biomechanical Energy Harvesting and Multifunctional Pressure Sensing. *Adv. Mater.* 30, 1–12 (2018).
6. Yang, P. et al. A Flexible, Stretchable and Shape-Adaptive Approach for Versatile Energy Conversion and Self-Powered Biomedical Monitoring. *Adv. Mater.* 27, 3817–3824 (2015).
7. Wang, H. et al. Self-Powered Inhomogeneous Strain Sensor Enabled Joint Motion and Three-Dimensional Muscle Sensing. *ACS Appl. Mater. Interfaces* 11, 34251–34257 (2019).
8. Zhang, C. et al. A stretchable dual-mode sensor array for multifunctional robotic electronic skin. *Nano Energy* 62, 164–170 (2019).
9. Homberg, B. S., Katzschmann, R. K., Dogar, M. R. & Rus, D. Haptic identification of objects using a modular soft robotic gripper. *IEEE Int. Conf. Intell. Robot. Syst.* 2015-Decem, 1698–1705 (2015).
10. Thuruthel, T. G., Shih, B., Laschi, C. & Tolley, M. T. Soft robot perception using embedded soft sensors and recurrent neural networks. *Sci. Robot.* 4, eaav1488 (2019).
11. Hughes, J. & Iida, F. Tactile Sensing Applied to the Universal Gripper Using Conductive Thermoplastic Elastomer. *5*, 512–526 (2018).
12. Liao, Z. et al. A tactile sensor translating texture and sliding motion information into electrical pulses. *Nanoscale* 7, 10801–10806 (2015).
13. Deng, W. et al. Cowpea-structured PVDF/ZnO nanofibers based flexible self-powered

- piezoelectric bending motion sensor towards remote control of gestures. *Nano Energy* 55, 516–525 (2019).
14. Peele, B. et al. Untethered Stretchable Displays for Tactile Interaction 1. 6, 142–149 (2019).
 15. Liang, G., Wang, Y., Mei, D., Xi, K. & Chen, Z. Flexible Capacitive Tactile Sensor Array with Truncated Pyramids as Dielectric Layer for Three-Axis Force Measurement. *J. Microelectromechanical Syst.* 24, 1510–1519 (2015).
 16. Ai, Y. et al. All rGO-on-PVDF-nanofibers based self-powered electronic skins. *Nano Energy* 35, 121–127 (2017).
 17. Li, G. & Zhu, R. A Multisensory Tactile System for Robotic Hands to Recognize Objects. *Adv. Mater. Technol.* 1900602, 1900602 (2019).
 18. Massari, L. et al. A Machine-Learning-Based Approach to Solve Both Contact Location and Force in Soft Material Tactile Sensors. *Soft Robot.* 00, 1–12 (2019)

Reviewers' comments (R2):

Reviewer #1 (Remarks to the Author):

I would like to thank the authors for addressing my comments and the comments of other reviewers, and extensively describing the changes they made. Their explanatory responses and manuscript additions in my concerns for the length of the paper and necessity of the digital twin experiment made the manuscript richer and more streamlined.

The machine learning experiment added to section "Data Processing via Machine Learning Technology" offers additional insight over the sensor performance in gripping tasks. The results seem to validate the usefulness of the two TENG sensors in grasping tasks of simple objects, with confusion matrices that reflect good performance. The authors need to describe the data collection and training procedure of the SVM: a) The authors claim that they used the voltages of electrodes and applied PCA to extract features, however they should mention exactly what features they have used as input to SVM (such as voltage peaks, RMS value etc.), and technical details on the SVM (kernel used etc). b) The authors need also to describe exactly how the grasping motion is performed to gather data for the 100 grasps on each object, and the finger contact location on the object. If the grasping motion is performed the same way as shown in Movie S5, there is a good chance that the 100 grasps for each object are similar to one another. The 80:20 split over all objects is also a factor for the good performance, as the usage of all objects in the training process ensures that the SVM recognises data very similar to the ones that have been trained upon. I understand that the authors argue their system is suited for applications where the robot encounters the same object (such as fruit picking), and

such a method of training is viable. I suggest the authors conduct an experiment where all data from 10-11 objects are used in training and all data from the rest 5-6 objects are used in testing. This will enable the research community understand the applicability of their sensors in more general object recognition, where generalisation to unknown objects is required.

Reviewer #2 (Remarks to the Author):

In the revised manuscript, the authors clearly addressed all the concerns I raised. It can be seen that the authors made a great effort to clarify all uncertainties and confusing points in the manuscript. It is an impressive engineering development of a TENG-based technology, while I still don't think the scientific impacts and novelty are very high. Nonetheless, the quality of work in this manuscript is high and can be recommended for publication on Nature Communications.

Reviewer #3 (Remarks to the Author):

The reviewer has reviewed the submission as well as the rebuttal to the original reviewer comments. The reviewer finds that the authors have effectively addressed the comments of the reviewers.

Manuscript number: NCOMMS-20-04528B

Title: Triboelectric Nanogenerator Sensors for Soft Robotics Aiming at Digital Twin Applications

Authors: Tao Jin, Zhongda Sun, Long Li, Quan Zhang, Minglu Zhu, Zixuan Zhang, Guangjie Yuan,

Tao Chen, Yingzhong Tian, Xuyan Hou and Chengkuo Lee

Responses to the reviewers:

Reviewer #1

Comment 1: The authors claim that they used the voltages of electrodes and applied PCA to extract features, however they should mention exactly what features they have used as input to SVM (such as voltage peaks, RMS value ea.), and technical details on the SVM (kernel used etc).

Response:

We thank a lot for reviewer's valuable comment. For the input of the PCA¹, we used the raw data of 15 channels of each grasp which is shown in Fig. R1 (Different columns show the time-domain signals of 15 channels of different objects during one grasping motion). The data

length for each channel is 200, so there are $200 * 15 = 3000$ features for each sample and each feature means one data point in the time domain during grasping. Then the PCA will extract features and reduce the dimensionality from 3000 to a few hundred, that means the final input to SVM² will be hundreds of data points of each sample which can best distinguish these objects. For better understanding, the enlarged voltage waveforms of each channel for grasping the banana are shown in Fig. R2, where the outputs of channel 1-9, 11, 13, 15 are from T-TENG sensors, and the outputs of 10, 12, 14 are from L-TENG sensors. All the data points of these channels will be directly used as input to the ML process. Due to the larger contact area when grasping the banana, channel 2, 3, 11 and 13 shows greater voltage value than other channels among T-TENG sensors, which can be further visualized as peak voltage maps as the reference to distinguish different objects by the human eye shown in Fig. R3. But the machine distinguishes these objects based on the hundreds of data points extracted from the 3000 data points of 15 channels for each sample, not just the voltage peaks, peak values or RMS values. The technical details of the SVM can be found in Table 1. We used the linear kernel and optimized the accuracy of the SVM according to commonly used parameters, and found that the optimal recognition accuracy of 98.125% can be achieved when the penalty parameter C is $1 * 10^{-2}$ and dimensionality of the data feature is reduced to 200 by PCA.

Actual Change: The concerned content has already been added and highlighted in Page 18, line 11-25 and Page 19, line 12-19. Fig. R1 and Table 1 in reply are provided in the Supplementary Materials (Fig. S14, Table S4).

Fig. R1. The input voltage signals of 15 channels for 16 different grasped objects.

Fig. R2. The input voltage signals of 15 channels for grasping the banana.

Table 1 Parameter optimization of SVM and PCA

Classification accuracy		Penalty parameter C			
		1×10^{-2}	1×10^{-1}	1	10
Linear kernel	PCs = 100	96.25%	95.938%	95.938%	95.938%
	PCs = 120	96.563%	96.563%	96.563%	96.563%
	PCs = 150	97.5%	97.5%	97.5%	97.5%
	PCs = 200	98.125%	97.813%	97.813%	97.813%
	PCs = 250	97.5%	97.5%	97.5%	97.5%

Comment 2: The authors need also to describe exactly how the grasping motion is performed to gather data for the 100 grasps on each object, and the finger contact location on the object. If the grasping motion is performed the same way as shown in Movie S5, there is a good chance that the 100 grasps for each object are similar to one another.

Response:

Thank you for reviewer's kind advice. In terms of the primary demonstration of showing the feasibility for object recognition in warehouse or factory where the robotic gripper usually grasps products with the same position and angle, we try to maintain a relatively fixed position for 100 grasps for each object during the data collection process to achieve a better prediction result with less samples. The peak voltage maps of the T-TENG sensors for different objects shown in Fig. R3

provide a reference for the finger contact location on the object. The peak voltages of the 12 T-TENG sensors for each sample are normalized between zero and one, and a darker color means larger contact area and contact force at this location during grasping. The grasping pressure differences among the 12 T-TENG sensors between objects are also clearly illustrated in this figure. Though the fabrication errors existing in our homemade sensors and pneumatic fingers may result in the asymmetrical distribution of the peak voltages in the T-TENG sensors for symmetric objects, this will not affect the prediction ability of the system because both the training and testing process are done by one specific gripper and the error between different sensors is also the same for all objects. This problem can be solved in the future by using more stringent and unified fabrication standards.

Additionally, for those grasps, some of them from different objects may look similar at a certain time frame (i.e., the time of taking photo, or the time of drawing the pressure maps). However, as we mentioned in main manuscript and above, we are using a period of output data (200 data points at time series) of 15 channels as a training sample, which means the data includes the information of the contact force, speed, sequences, contact positions, latency, and the contact durations etc. Those multi-dimensional features, rather than the individual grasping position, will then define the identity of the grasped object. As a result, as long as those objects have distinct shape differences, the object recognition can be achieved.

Actual Change: We have added the detailed explanation in Text S5 in Supplementary Materials. Fig. R3 has also been added in the Supplementary Materials as Fig. S17.

Fig. R3. Peak voltage maps of the T-TENG sensors for 16 different objects.

Comment 3: I suggest the authors conduct an experiment where all data from 10-11 objects are used in training and all data from the rest 5-6 objects are used in testing. This will enable the research community understand the applicability of their sensors in more general object recognition, where generalisation to unknown objects is required.

Response:

We appreciate the reviewer’s suggestion on the experiment of generalisation to unknown objects. Here we choose tennis ball, big box and short can to represent the three most common shapes of objects in daily life: sphere, cube and cylinder, as the unknown objects for testing. In the experiment, we still use the 100 grasp data of the rest 13 objects for training, therefore each raw data consists a period of output data in time series of 15 channels, 200 data points * 15 channels = 3000 features in total for each sample as the input to the machine. Again, we understand that the machine learning algorithm identifies objects still based on a complete set of 15 channel sensor data in each grasp, i.e., hundreds of data points extracted from the 3000 data points of 15 channels by PCA for each sample data, not just the peak values.

For human being, it could be relatively easy to tell the difference when we look at the peak voltage map. Therefore, we conduct the next three experiments to verify effectiveness of general object recognition, in particular, the generalisation to unknown objects, under our sensor design and

present machine learning algorithm.

(1) Tennis ball test

The first experiment is to see the recognition outcome from the machine for the unknown object like a tennis ball based on the previous training data which does not have tennis grasp data at all. We collected 100 grasp data samples for the tennis ball, i.e., the unknown object data for the trained machine model. Then we see the recognition outcomes for these 100 grasp data samples for the tennis ball, 51 samples are recognized as the baseball, and 40 samples are predicted as the apple. From the 51 samples, one of the sample data is selected and only the peak voltage value is identified and provided as the peak voltage map of the T-TENG sensors of tennis ball in the Fig. R4a. Then, among the 40 samples, one of the sample data is selected and the peak voltage map of the T-TENG sensors of tennis ball is identified as shown in the Fig. R4b. One of the typical training data of the baseball and the apple are selected and the peak voltage maps are shown in Fig. R4c and Fig. R4d. We try to leverage the peak voltage map to help us to figure out any relevant information for the machine to do the unknown object recognition.

The values and distribution of sensor outputs among channel 1-3, 7-9, 11 and 13 are quite similar among these three spherical objects. Furthermore, the output of channel 15 in Fig. R4a is close to the value in the Fig R4c, while the output of channel 15 in Fig. R4b is close to the value in the Fig R4d. This could be a factor observed by the machine and then suggest the sample data of R4a as the baseball, and the sample data of R4b as the apple.

Back to the point indicated by reviewer “This will enable the research community understand the applicability of their sensors in more general object recognition, where generalisation to unknown objects is required.”, I think the above results show that our sensors and machine algorithm works effectively to recognize the spherical object.

Fig. R4. Peak voltage maps of tennis ball samples predicted as a) the baseball and b) the apple respectively; peak voltage maps of c) the typical baseball sample and d) the typical apple sample.

(2) Big box test

The second unknown object recognition experiment is to test the big box grasp data for 100 times based on the trained model as same as the experiment 1. For the big box, most samples (total 70)

are recognized as the orange, and only a few samples (21) are predicted as the baseball. From the 70 samples, one of the sample data is selected and the peak voltage map of the T-TENG sensors of big box is provided in the Fig. R5a. Then, among the 21 samples, one of the sample data is selected and the peak voltage map of the T-TENG sensors of big box is identified as shown in the Fig. R5b. One of the typical training data of the orange and the baseball are selected and the peak voltage maps are shown in Fig. R5c and Fig. R5d.

Although we have one cube shaped object, i.e, the small box, in the trained model, we do not get the recognition outcome of recognizing “big box” as the “small box”. The dimension of big box and small box are 10 cm * 4 cm * 14cm and 8 cm * 2 cm * 14cm, respectively, where the length of the big box is slightly longer than that of the small box, and the width of the big box is twice that of the small box. When we look at the 7 and 8 voltage maps in the Fig R3, the voltage value in channel 11 for the small box is obviously the highest among all channels, which is quite different from that of the big box where the voltage value in channel 11 is much smaller than that of the channel 9. The voltage distribution difference between the small box and the big box may finally make the machine recognize the big box as other existing objects in the training set with more similar voltage distribution.

On the other hand, when we look at the peak voltage maps in Fig R5, the values and distribution of sensor outputs among channel 1-3 and channel 11 of the big box shown in the Fig. R5a are quite similar to that of the orange shown in Fig. R5c. This could be a factor that most big boxes with such voltage distribution are identified as the orange. But for a few big box samples which have higher voltage values in channel 2-3 and 11 shown in the Fig. R5b, the baseball that also has higher values in these channels (Fig. R5d) than that of the orange becomes the most possible prediction result of these samples.

After all, with the general understanding of machine learning, insufficient trained data in the cubic objects may make the machine suggest the 2nd best-matched objects (the orange) existing in the training set and putting more cubic objects of different sizes and types into the training set may be helpful to enhance the machine’s ability to recognize the general cubic objects.

Fig. R5. Peak voltage maps of big box samples predicted as a) the orange and b) the baseball respectively; peak voltage maps of c) the typical orange sample and d) the typical baseball sample.

(3) Short can test

The third experiment is to identify the unknown short can grasp data for 100 times based on the trained model. Among these samples, more than half (52) are predicted as the long can, and 23 and 22 samples are recognized as the orange and the banana respectively. Among the 52, 23 and 22 samples, one of the sample data is selected from each category and the corresponding three peak voltage maps of the T-TENG sensors of the short can are provided in the Fig. R6a, Fig. R6b and Fig. R6c respectively. One of the typical training data of the long can, orange and banana are selected and the peak voltage maps are shown in Fig. R6d, Fig. R6e and Fig. R6f.

According to the peak voltage maps in Fig. R6a and Fig. R6d, it is clear that the short can and the long can have almost the same peak voltage distribution among the 12 T-TENG sensors, where peak voltage values in channel 1-3, channel 9, channel 11, channel 13 and channel 15 are usually larger than the remaining channels and the range is also relatively stable, which may make the recognition result of the long can dominates. For the few samples of short can that have relatively low values in channel 4-6 shown in the Fig. R6c, the typical banana sample which also has small peak voltages in these channels in the Fig. R6f shows a higher similarity to these samples and this may be the decisive factor that the sample data of Fig. R6c is recognized as the banana. However, for the sample data visualized in Fig. R6b and Fig. R6e, it's difficult to directly tell the similarity just based on the peak voltage maps and the machine may make the prediction based on more features with higher dimensionality in the raw data, not just the peak voltage values.

The above results show that our sensors and machine algorithm also work effectively to recognize the cylindrical object even though there are only a small number of cylindrical samples in the training set.

Fig. R6. Peak voltage maps of short can samples predicted as a) the long can, b) the orange and c) the banana respectively; peak voltage maps of d) the typical long can sample, e) the typical orange sample and f) the typical banana sample.

In summary, according to the prediction results of unknown objects given by the pre-trained SVM model based on the raw data (3000 data points) of samples, as well as the referential analysis of the voltage maps, the above results for three unknown objects indicate that our developed sensors have a certain recognition ability when encountering general spherical and cylindrical objects,

which will be highly likely to be identified as objects with similar shapes in the training data set. While in the experimental 2, due to insufficient trained data in the cubic objects, we realized that machine may suggest the 2nd best-matched results because of the relative values and distribution observed in the outputs of the 12 T-TENG sensors. The prediction ability of our system for general cubic objects can be enhanced in the future if we put more cubic objects of different sizes and types into the training set and make the machine learn more unique features of cubic objects.

It is worth to mention that, SVM technique as a supervised learning algorithm, all of the recognizable objects were labelled during the training process. Hence, if we introduce the unknown object for testing without training, then SVM will not be able to provide the true identity of itself (since it does not exist), but to categorized it into the trained labels which have the most similar features. In another word, for unknown objects, the function of SVM-based finger grasping becomes classification (based on the existed classes, i.e., the trained 10~11 objects), rather than the precise recognition (i.e., provide the true identity of itself), and the classification should be shape, position, and size-based recognition. Hence, that is the reason for above test results of the unknown objects recognition, i.e., the tennis ball is categorized into the baseball or the apple, and the short can is identified as the long can. Moreover, it is universally accepted that the performance of generalization also relies on the population (classes) of trained data set, and hence, for larger database, there will be more chance to find the similar object which matches the unknown object better. For the above example of big box test, if we can establish a data base with more trained objects which contains other boxes with different sizes or shapes, the recognition performance will then be effectively improved.

Reviewer #2

Comments:

In the revised manuscript, the authors clearly addressed all the concerns I raised. It can be seen that the authors made a great effort to clarify all uncertainties and confusing points in the manuscript. It is an impressive engineering development of a TENG-based technology, while I still don't think the scientific impacts and novelty are very high. Nonetheless, the quality of work in this manuscript is high and can be recommended for publication on Nature Communications.

Response:

Thank you very much for your positive comments on the manuscript.

Reviewer #3

The reviewer has reviewed the submission as well as the rebuttal to the original reviewer comments. The reviewer finds that the authors have effectively addressed the comments of the reviewers.

Response:

Thank you very much for your positive comments on the manuscript.

Reference:

1. Abdi, H. & Williams, L. J. Principal component analysis. *WIREs Comp Stat* 2, 433–459 (2010).
2. Burges, C. J. C. A Tutorial on Support Vector Machines for Pattern Recognition. *Data*

Mining and Knowledge Discovery 2, 121–167 (1998).

Reviewers' comments (R3):

REVIEWERS' COMMENTS

Reviewer #1 (Remarks to the Author):

I would like to thank again the authors for their detailed experiments, and for analytically addressing my comments.

The PCA and SVM section is now more complete and helps the reader understand how the data were gathered, processed, and utilised.

The new experiments are important because they demonstrate both the strengths and the limitations of the TENG sensor in object recognition. The TENG signals are similar for rounded surfaces, leading to good recognition of curved surfaces (sphere and cylinder recognition experiment), and additional research can improve the results in flat surface recognition.

I suggest publication of the article, and I wish the authors further success in their future research.

Manuscript number: NCOMMS-20-04528B

Title: Triboelectric Nanogenerator Sensors for Soft Robotics Aiming at Digital Twin Applications

Authors: Tao Jin, Zhongda Sun, Long Li, Quan Zhang, Minglu Zhu, Zixuan Zhang, Guangjie Yuan,

Tao Chen, Yingzhong Tian, Xuyan Hou and Chengkuo Lee

Responses to the reviewers:

Reviewer #1

Comments:

I would like to thank again the authors for their detailed experiments, and for analytically addressing my comments.

The PCA and SVM section is now more complete and helps the reader understand how the data were gathered, processed, and utilised.

The new experiments are important because they demonstrate both the strengths and the limitations of the TENG sensor in object recognition. The TENG signals are similar for rounded surfaces, leading to good recognition of curved surfaces (sphere and cylinder recognition experiment), and additional research can improve the results in flat surface recognition.

I suggest publication of the article, and I wish the authors further success in their future research.

Response:

Thank you very much for your positive comments on the manuscript.